# Implicit Action Chunking for Smooth Continuous Control

**Bosun Liang** [1]  **Shuo Pei** [1]  **Zirui Chen** [2]  **Chuanzhi Fan** [2]  **Chen Sun** [1]  **Yuankai Wu** [3]  **Huachun Tan** [2]
**Yong Wang*** [1]

## Abstract

Reinforcement learning often produces high-frequency oscillatory control signals that undermine the safety and stability required for physical deployment. Explicit action chunking addresses this by predicting fixed-horizon trajectories but scales the policy output dimension proportionally with the horizon length, leading to optimization difficulties and incompatibility with standard step-wise interaction. To overcome these challenges, this paper proposes Dual-Window Smoothing (DWS), an implicit action chunking framework for smooth continuous control. Unlike explicit methods, DWS enforces temporal coherence without expanding the action space. It uses a dual-window design: an execution window that ensures physical smoothness through deterministic modulation, and a value window that aligns temporal-difference targets over the horizon to correct critic bias caused by open-loop execution. DWS also includes a lightweight actor-side temporal regularizer based on first-order action differences to promote global continuity. This design effectively bridges the gap between temporal abstraction and reactive step-wise control. Experiments on benchmarks including the DeepMind Control Suite and industrial energy management tasks show that DWS outperforms state-of-the-art (SOTA) baselines. In complex vision-based autonomous driving tasks, DWS achieves smoother control, safer behavior with reduced jitter, and attains a 100% success rate.

[1]Department of Data and Systems Engineering, The University of Hong Kong, Hong Kong SAR, China [2]Beijing Institute of Technology, Zhuhai, China [3]College of Computer Science, Sichuan University, Chengdu, China. Correspondence to: Yong Wang <wy0304@hku.hk>.

*Proceedings of the $43^{rd}$ International Conference on Machine Learning*, Seoul, South Korea. PMLR 306, 2026. Copyright 2026 by the author(s).

## 1. Introduction

Deep Reinforcement Learning (DRL) has achieved significant milestones in complex continuous control, from high-dimensional robotic manipulation to autonomous driving (Elguea-Aguinaco et al., 2023; Luo et al., 2025; Wu et al., 2024b; Xu et al., 2025). Despite these capabilities, physical deployment imposes a strict requirement for *temporal coherence*, defined as the generation of smooth and predictable trajectories that respect actuator limits (Mysore et al., 2021; Shen et al., 2020). However, a fundamental structural mismatch exists because standard DRL algorithms optimize decisions in a step-wise manner, whereas physical hardware demands temporally smooth actuation to manage inertia and bandwidth limitations (Mysore et al., 2021; Dosovitskiy et al., 2017). Consequently, learned policies often exhibit high-frequency oscillations that are difficult for physical actuators to execute, compromising system reliability (Tassa et al., 2018). Such instability is prohibitive in dynamic scenarios like motion control, where control smoothness is a prerequisite for system safety.

Existing research addresses control smoothness primarily through explicit regularization or structured policy parameterizations. Regularization-based methods penalize temporal action derivatives to mitigate high-frequency fluctuations, but may limit policy expressiveness (Mysore et al., 2021; Shen et al., 2020). In parallel, architectural approaches seek intrinsic stability by enforcing Lipschitz continuity, exemplified by methods like L2C2 and MLP-SN (Kobayashi, 2022; Wang et al., 2024a). LipsNet and LipsNet++ further combine adaptive Lipschitz bounds with learnable frequency-domain filters to handle observation noise (Song et al., 2023; 2025), while SmODE uses ODE-based mechanisms to regulate trajectory dynamics (Wang et al., 2025a). Despite these architectural enhancements, such approaches remain bound to a step-wise decision paradigm that infers actions based solely on instantaneous states. This reliance restricts the capacity to model long-horizon temporal dependencies and necessitates a compromise between maximizing returns and satisfying smoothness constraints, particularly in high-dimensional perception scenarios where systematic validation remains limited.

To transcend step-wise limitations, recent work has explored

Action Chunking (Bharadhwaj et al., 2024), which redefines the control unit as a fixed-horizon sequence. This paradigm projects actions onto a trajectory manifold, enforcing intra-chunk coherence while mitigating the non-Markovian stochasticity of instantaneous decisions (Black et al., 2025a;b). In robotic imitation and Vision-Language-Action (VLA) models (Intelligence et al., 2024; Chen et al., 2024), these open-loop chunks ensure spatiotemporal continuity and enable inference-time refinement to stabilize execution without retraining. Extending this temporal abstraction to RL, algorithms such as Q-Chunking reformulate the MDP to operate over action sequences,leveraging multi-step Bellman backups to accelerate value propagation and bridge the credit assignment gap in long-horizon tasks (Li et al., 2025b). Subsequent advancements like Decoupled Q-Chunking further refine this mechanism by distinguishing the planning horizon of the actor from that of the critic, thereby mitigating open-loop execution errors while maintaining training efficiency (Li et al., 2025a). These studies show that short-term temporal commitment is a useful inductive bias for stabilizing high-frequency control and facilitating exploration.

However, these methods fall under the paradigm of explicit action chunking, which necessitates the direct prediction of fixed-horizon trajectory segments. This approach introduces three problems. (1) It offers limited guarantees on execution-level smoothness, the open-loop nature of chunk execution allows for rapid intra-chunk variance and induces sharp discontinuities at replanning boundaries (Yang et al., 2025). (2) It exacerbates optimization complexity by expanding the policy output from $\mathbb{R}^d$ to $\mathbb{R}^{hd}$, which can hamper exploration efficiency and training stability as the horizon $h$ grows. (3) Explicit chunks create an architectural incompatibility with standard step-wise interaction interfaces, complicating integration with established actor-critic backbones and experience replay pipelines. Crucially, this structural mismatch hinders effective integration with expert-guided DRL, since conventional safety shielding and fallback mechanisms depend on instantaneous, point-wise action overrides that conflict with fixed-horizon action chunks. *These limitations motivate a pivotal inquiry: rather than explicitly inflating the action space, can we capture the stabilizing benefits of temporal abstraction implicitly, achieving superior smoothness while retaining the reactive feedback and modularity of step-wise control?*

To address these challenges, we propose **Dual-Window Smoothing (DWS)**, an implicit action chunking framework that bridges the gap between temporal abstraction and step-wise control by coupling an $h$-step execution window with a synchronized value window. Diverging from explicit trajectory prediction, DWS retains a standard $d$-dimensional action output to circumvent optimization instability, instead achieving smoothness via an execution operator that trans-

forms reference actions into locally coherent motion, reinforced by temporal regularization. Structurally, this design preserves the established actor–critic interface and standard transition-based experience replay, while an auxiliary on-policy window buffer aligns critic training with the $h$-step execution commitment. Consequently, DWS functions as a plug-and-play module for common DRL backbones and maintains full compatibility with expert-guided DRL, as the executed action stream supports instantaneous, point-wise safety interventions without disrupting the windowed architecture. These attributes endow DWS with significant potential for complex real-world control tasks.

Our contributions are threefold: (1) We introduce the DWS framework, which enables implicit action chunking within standard step-wise interfaces by coupling a deterministic Execution Window with lightweight boundary regularization. This design enforces execution-level smoothness and temporal coherence without expanding the action manifold or sacrificing high-frequency reactivity. (2) We propose a horizon-aligned value learning mechanism via a synchronized Value Window, which constructs terminal-safe, window-consistent TD targets from contiguous executed segments. This effectively resolves the misalignment between step-wise value estimation and temporally committed execution, mitigating critic myopia. (3) We demonstrate superior performance across diverse domains, from vector-based control and real-world energy management to high-dimensional visual autonomous driving. DWS consistently outperforms SOTA baselines and exhibits seamless compatibility with expert-guided DRL, notably achieving a 100% success rate in safety-critical overtaking scenarios.

## 2. Preliminaries

**Actor-Critic Backbone.** We formulate the continuous control problem as a Markov Decision Process (MDP) defined by $(\mathcal{S}, \mathcal{A}, P, r, \gamma)$, where $\mathcal{S}$ and $\mathcal{A} \subset \mathbb{R}^d$ denote the state and action spaces. The objective is to maximize the expected discounted return $J(\phi) = \mathbb{E}[\sum_{t=0}^{\infty} \gamma^t r(\mathbf{s}_t, \mathbf{a}_t)]$. Prominent off-policy algorithms, such as TD3 (Fujimoto et al., 2018) and SAC (Haarnoja et al., 2018), adopt an actor-critic architecture comprising a policy $\pi_\phi$ and a value function $Q_\theta$. The critic parameters $\theta$ are updated by minimizing the Bellman error against a temporal-difference (TD) target $y_t = r_t + \gamma(1 - d_t)Q_{\theta^-}(\mathbf{s}_{t+1}, \tilde{\mathbf{a}}_{t+1})$, where $\theta^-$ denotes a slowly updating target network. Concurrently, the actor $\pi_\phi$ is optimized to maximize the estimated return $Q_\theta(\mathbf{s}, \pi_\phi(\mathbf{s}))$. Crucially, these standard approaches perform optimization in a point-wise manner, determining actions based solely on instantaneous state evaluations. This absence of explicit temporal modeling frequently leads to high-frequency oscillations and control instability.

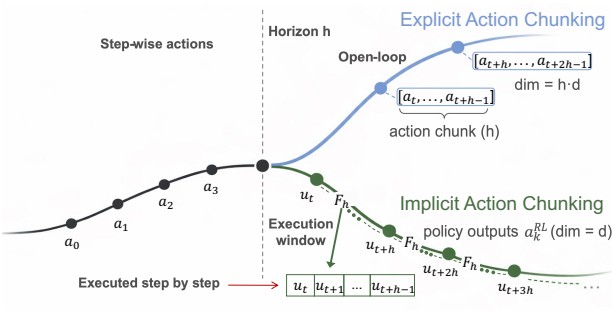

*Figure 1.* **Explicit vs. implicit action chunking.** Explicit chunking outputs an $h$-step action sequence in one decision (dimension $hd$) and executes it open-loop. DWS keeps a $d$-dimensional policy output $a$ and induces chunk-like temporal coherence by executing $u = \mathcal{F}_h(a)$ step-wise.

**Expert-guided Reinforcement Learning.** In safety-critical domains or tasks with sparse rewards, relying solely on stochastic exploration is often inefficient or hazardous. Expert-guided RL (EGRL) mitigates these issues by integrating an external authority $\pi^E$ directly into the learning loop. Such an expert can be derived from various sources, including human demonstrations (Wu et al., 2024c), optimal controllers (Liu et al., 2025), or script-based planners that are queried interactively during training (Pan et al., 2025). Within this widely adopted paradigm, expert data is typically utilized to accelerate convergence via behavioral cloning regularizers or to guarantee a performance lower bound during the initial training stages (Cao et al., 2023). Formally, the data collection process is modulated by a binary intervention signal $g_t \in \{0, 1\}$. At each discrete time step $t$, the system either operates autonomously or defers to the expert, resulting in a composite behavior policy $\beta$:

$$\mathbf{a}_t = \begin{cases} \mathbf{a}_t^{RL} \sim \pi_\phi(\cdot \mid \mathbf{s}_t) & \text{if } g_t = 0 \quad \text{(Agent)}, \\ \mathbf{a}_t^{E} \sim \pi^E(\cdot \mid \mathbf{s}_t) & \text{if } g_t = 1 \quad \text{(Expert)}. \end{cases}$$

where $\mathbf{a}_t^{RL}$ is typically the deterministic output $\pi_\phi(\mathbf{s}_t)$ during deployment or evaluation. To effectively distill expert knowledge while retaining the capacity for self-improvement, the optimization objective combines standard return maximization with an adaptive imitation term.

$$\mathcal{L}_{\text{EG}}(\phi) = \mathbb{E}\left[ -Q_\theta(\mathbf{s}_t, \pi_\phi(\mathbf{s}_t)) + \lambda \cdot g_t \cdot \omega_t \cdot \left\| \mathbf{a}_t^{E} - \pi_\phi(\mathbf{s}_t) \right\|_2^2 \right]$$

The weighting term $\omega_t \geq 0$ regulates the supervision strength based on the critic's evaluation, typically defined as the advantage of the expert action over the agent's policy.

## 3. Method

### 3.1. Implicit Action Chunking Formulation

Standard explicit action chunking expands the policy output space to $\mathbb{R}^{h \times d}$, directly predicting fixed-horizon (h)

action trajectories, as illustrated in Figure 1. In contrast, we formalize implicit action chunking as a generative process that decouples high-level decision-making from low-level execution dynamics. Specifically, the policy $\pi_\theta(a \mid s)$ retains a standard low-dimensional output space $a \in \mathbb{R}^d$, where the action serves as a latent control variable. Temporal coherence is induced structurally through a deterministic execution operator $\mathcal{F}_h : \mathbb{R}^d \to \mathbb{R}^{h \times d}$, which maps an atomic action $a_t$ into a smooth trajectory segment $\mathbf{u}_{t:t+h-1} = \mathcal{F}_h(a_t)$. This formulation restricts the agent's search space to a smooth submanifold embedded in the full sequence space $\mathbb{R}^{h \times d}$, thereby implicitly regularizing the optimization landscape and promoting temporally coherent control. Formally, DWS optimizes the expected return over trajectories generated by this implicit process:

$$\max_{\pi_\theta} \quad \mathbb{E}_\tau \left[ \sum_{t=0}^{T} \gamma^t r(s_t, u_t) \right] \tag{1}$$
$$\text{s.t.} \quad \mathbf{u}_{k:k+h-1} = \mathcal{F}_h(a_k \sim \pi_\theta(\cdot \mid s_k)).$$

where $k = mh \ (m \geq 0)$ indexes window boundaries. At each boundary $k$, the policy samples $a_k \sim \pi_\theta(\cdot \mid s_k)$ and the execution operator generates $\mathbf{u}_{k:k+h-1} = \mathcal{F}_h(a_k)$, which is applied step-by-step as $u_t$ at environment step $t$.

To operationalize Eq. (1) within standard actor–critic backbones, DWS adopts a dual-window design. The **Execution Window** applies $\mathcal{F}_h$ online to generate step-wise executed actions under an $h$-step commitment. The **Value Window** then trains the critic with window-aligned $h$-step supervision computed from contiguous executed segments, ensuring value learning matches the same execution horizon.

### 3.2. Execution Window

To physically suppress high-frequency oscillations, DWS instantiates the abstract operator $\mathcal{F}_h$ as a deterministic temporal modulation kernel. This mechanism decouples the decision frequency (controlled by $\pi$) from the actuation frequency (controlled by the environment). Formally, at the initiation of a window at time $t$, the policy generates a latent reference action $a_t \sim \pi(\cdot \mid s_t)$. The physical action sequence $\mathbf{u}_{t:t+h-1}$ is then generated via a modulation profile $\mathbf{w} \in \mathbb{R}^h$, as illustrated in Figure 2:

$$u_{t+k} = w_k \cdot a_t, \quad \forall k \in \{0, \dots, h-1\}, \tag{2}$$

where $w_k$ is a scalar weight governing intra-window dynamics. We investigate two distinct spectral profiles for $\mathbf{w}$: (i) Zero-Order Hold (ZOH): Defined as $w_k = 1, \forall k$. This profile enforces rigid temporal consistency, effectively acting as a low-pass filter that strictly caps the control frequency at $1/h$ to maximize trajectory correlation. (ii) Dissipative Decay: Defined by a monotone sequence

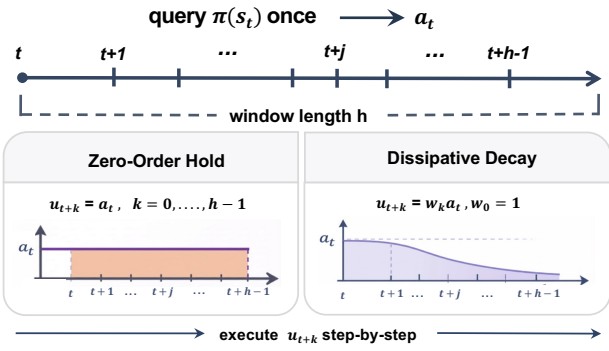

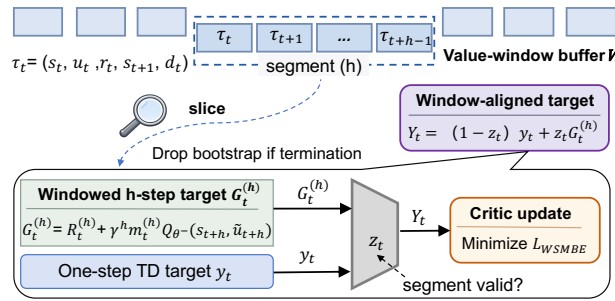

*Figure 2.* **Execution Window.** At each window boundary, the policy outputs a reference action $a_t$, and the execution profile **w** produces step-wise executed actions $u_{t:t+h-1}$ via $u_{t+k} = w_k a_t$. ZOH yields constant within-window execution, while Dissipative Decay gradually attenuates the magnitude.

*Figure 3.* **Value Window.** A contiguous length-$h$ executed segment is sliced from the ordered buffer $\mathcal{W}$ to form a $h$-step windowed target $G_t^{(h)}$. The critic supervision target interpolates between the one-step TD target and the windowed target using the segment-valid gate $z_t$.

$1 = w_0 \geq w_1 \geq \cdots \geq w_{h-1} \geq 0$. This profile attenuates the action magnitude within the open-loop interval, providing a simple actuation damping effect that reduces sensitivity to within-window drift and disturbances. In our implementation, we use a linear instantiation $w_k = 1 - \frac{k}{h}$, which yields $u_{t+k} = w_k a_t$ and smoothly decays the executed action to $w_{h-1} = 1/h$ at the end of the window.

**Proposition 1** (Intra-Window Smoothness). *Assume the policy output is bounded such that $\|a_t\| \leq A_{\max}$. For a given execution profile* **w**, *the discrete variation of the executed action $u$ within a window is uniformly bounded as*

$$\|\Delta u_{t+k}\| \leq |w_k - w_{k-1}| \cdot A_{\max}, \quad \forall k \in \{1, \ldots, h-1\}. \tag{3}$$

*Proof.* See Proposition A.1 in Section A.

**Remark 1.** *For the zero-order hold profile, $|w_k - w_{k-1}| = 0$, which yields perfect intra-window smoothness, i.e., zero discrete jitter. For the dissipative profile, the degree of smoothness is explicitly controlled by the decay rate of* **w**. *As a result, high-frequency action fluctuations are structurally suppressed within each window, independent of the stochasticity of the policy $\pi$.*

In this paper, we adopt a short window length ($h = 3$), ensuring that the policy is queried at a high frequency across window boundaries. This choice is supported by prior research, which demonstrates that a smaller action chunk length yields superior performance (Li et al., 2025b).

### 3.3. Value Window

A critical challenge in smoothing is the *training–execution mismatch*. Standard critics trained with one-step TD targets implicitly assume the policy can react immediately at $t+1$ (e.g., $r_t + \gamma Q(s_{t+1}, \tilde{u}_{t+1})$). When execution is windowed via a length-$h$ profile, this assumption is violated, which

can bias value estimates toward step-wise (potentially jittery) corrections and lead to critic myopia with respect to segment-wise coherent execution. Accordingly, value propagation should use an $h$-step *window-aligned* backup that matches the executed commitment.

To align valuation with execution under the same horizon $h$, we introduce the value window, as illustrated in Figure 3. During interaction, we maintain a lightweight ordered on-policy window buffer $\mathcal{W}$ that stores executed one-step transitions $(\mathbf{s}_t, \mathbf{u}_t, r_t, \mathbf{s}_{t+1}, d_t)$ in temporal order, where $d_t \in \{0, 1\}$ indicates termination. Unlike shuffled replay, $\mathcal{W}$ enables sampling contiguous length-$h$ segments required for windowed backups.

**Windowed $h$-step target.** From a contiguous executed segment in $\mathcal{W}$ starting at time $t$, we define the windowed return

$$G_t^{(h)} = \sum_{k=0}^{h-1} \gamma^k r_{t+k} + \gamma^h m_t^{(h)} Q_{\theta^-}(\mathbf{s}_{t+h}, \tilde{\mathbf{u}}_{t+h}), \tag{4}$$

where $\tilde{\mathbf{u}}_{t+h}$ is the bootstrap action from the target actor (with target noise if applicable), and the mask

$$m_t^{(h)} = \mathbf{1}\left[\sum_{k=0}^{h-1} d_{t+k} = 0\right] \tag{5}$$

avoids bootstrapping across termination. For twin critics, we use $Q_{\theta^-} = \min(Q_{\theta_1^-}, Q_{\theta_2^-})$.

**Window–step mixed Bellman error.** Let $y_t = r_t + \gamma(1 - d_t) Q_{\theta^-}(\mathbf{s}_{t+1}, \tilde{\mathbf{u}}_{t+1})$ denote the standard one-step TD target from replay, and let $z_t \in \{0, 1\}$ indicate whether a valid length-$h$ contiguous window (not crossing termination) is available in $\mathcal{W}$. We define the *window-aligned* target

$$Y_t = (1 - z_t) y_t + z_t G_t^{(h)}, \tag{6}$$

and train the critic by minimizing the **Window–Step Mixed Bellman Error**

$$\mathcal{L}_{\mathrm{WSMBE}}(\theta) = \mathbb{E}\Big[\big(Q_\theta(\mathbf{s}_t, \mathbf{u}_t) - Y_t\big)^2\Big]. \qquad (7)$$

This objective reduces to standard one-step TD at episode boundaries ($z_t = 0$), and otherwise enforces $h$-step supervision aligned with the executed commitment ($z_t = 1$), thereby coupling value propagation to the same horizon used by the execution window.

**Proposition 2** (Operator-Consistent Windowed Target). *Fix the execution operator $\mathcal{F}_h$ and consider the induced executed interaction as Markovian in an augmented state space that includes the within-window phase and the cached reference action. For any target critic $Q_{\theta^-}$, whenever a valid contiguous executed segment of length $h$ is available (i.e., $z_t = 1$), the windowed target $G_t^{(h)}$ defined in Eq. (4) constitutes an unbiased empirical sample of the corresponding $h$-step Bellman backup under the executed process:*

$$\mathbb{E}\Big[G_t^{(h)} \mid s_t, u_t\Big] = \big(\mathcal{T}_h^{\pi^{\mathrm{exec}}} Q_{\theta^-}\big)(s_t, u_t), \qquad (8)$$

*where $\mathcal{T}_h^{\pi^{\mathrm{exec}}}$ denotes the $h$-step policy-evaluation operator aligned with the same execution commitment. Bootstrapping is masked by $m_t^{(h)}$ as specified in Eq. (5).*

**Remark 2.** *As a consequence, minimizing the window-step mixed Bellman error in Eq. (7) performs Bellman regression toward the window-aligned evaluation operator whenever $z_t = 1$. When a full execution segment is unavailable, such as near episode boundaries, the objective naturally falls back to standard one-step temporal-difference learning ($z_t = 0$). A formal augmented-Markov characterization of the executed process and the induced operator is provided in Section A.*

### 3.4. Temporal Smoothness Regularizer

The execution window guarantees smoothness *within* a window, but the policy can still introduce discontinuities at window boundaries (e.g., between $t + h - 1$ and $t + h$). To encourage global continuity, DWS adds a lightweight actor-side temporal regularizer based on first-order action differences. Given the base actor loss $L_{\mathrm{base}}(\phi)$ from the backbone, we optimize

$$L_\pi(\phi) = L_{\mathrm{base}}(\phi) + \lambda_S \, \mathbb{E}\Big[\|\pi_\phi(\mathbf{s}_t) - \pi_\phi(\mathbf{s}_{t-1})\|_2^2\Big], \quad (9)$$

where $\lambda_S > 0$ is a scalar weight. We estimate the expectation using temporally adjacent, non-terminal state pairs sampled from $\mathcal{W}$ (avoiding cross-episode pairs), so the regularizer reflects the current interaction distribution induced by the execution window. Intuitively, this term softly regularizes the policy's discrete-time variation; under windowed execution, it primarily suppresses discontinuities at window boundaries and complements the hard intra-window coherence induced by Eq. (2).

**Backbone-agnostic instantiation.** DWS is a lightweight wrapper for off-policy actor–critic methods: we only (i) execute actions through the execution window profile (Eq. (2)) and (ii) train critics with the window-aligned target in Eq. (7), optionally adding the actor regularizer in Eq. (9). We keep the original Actor-Critic backbone objectives and target-network machinery unchanged. Full pseudocode is in Section A.2.

## 4. Experiments

### 4.1. Experimental Setup

**Environments.** We evaluate **DWS** across three control paradigms: (i) **DeepMind Control Suite (DMC)** (Tassa et al., 2018) with five tasks (Reacher-Easy, Reacher-Hard, Ball in Cup-Catch, Cart-pole-Swingup, Point Mass-easy); (ii) an industrial **electric-vehicle energy management (EMS)** (Wang et al., 2025b) task with vector observations, requiring smooth power split to reduce cost and wear; and (iii) **CARLA** (Dosovitskiy et al., 2017) end-to-end autonomous driving with two vision-based safety-critical scenarios (Overtaking, Braking). Full environment specifications are in Section D.1.

**Baselines.** We compare against standard off-policy actor–critic backbones (**TD3** (Fujimoto et al., 2018), **SAC** (Haarnoja et al., 2018)), representative smooth-control methods (**L2C2** (Kobayashi, 2022), **LipsNet++** (Song et al., 2025), **SmODE** (Wang et al., 2025a)), and an explicit temporal abstraction baseline **ActionChunk** (Li et al., 2025b). Baseline details are provided in Section B, and hyperparameters are provided in Section C.

**Metrics.** The primary metric for control smoothness is the **Average Fluctuation Rate (AFR)** (Song et al., 2025), which quantifies the temporal variance of the action sequence. Task-specific performance is measured by cumulative rewards (DMC), operational costs (EMS), and collision-free success rates (CARLA). Detailed metric formulations are deferred to Section D.2, Section F.2.4, and Section F.3.3.

### 4.2. Case Study of DeepMind Control Suite

To verify the proposed method, we evaluated it on five Deep-Mind tasks representing distinct physical control challenges. Figure 4 presents the asymptotic performance, where DWS (Blue) achieves the largest coverage area for both TD3 and SAC backbones, indicating the best trade-off between control smoothness and task return. Quantitative results confirm that DWS consistently outperforms baselines by avoiding their structural pitfalls. Specifically, explicit chunking (ActionChunk) succumbs to the curse of dimensionality in impulse-driven tasks, failing to converge in Ball-in-Cup (Return: 3.12). Similarly, constrained smoothers like Lip-

sNet++ suffer from exploration collapse in precision tasks, yielding a poor return of 0.40 ±1.05 in Reacher-Hard. In contrast, DWS leverages implicit action chunking to maintain efficient exploration on the low-dimensional manifold, achieving near-optimal returns of 975.35 in Reacher-Hard and 966.95 in Ball-in-Cup, demonstrating superior robustness across diverse dynamics.

We further analyze the source of this performance gain by examining control smoothness and temporal action structure. Figure 5 presents a microscopic view of action trajectories in Reacher-Hard. Standard RL baselines tend to degenerate into saturation-level switching behaviors to minimize instantaneous tracking error, resulting in pronounced high-frequency oscillations. In contrast, DWS produces very smooth and stable control actions by generating locally coherent signals through the execution window design. This design uses a deterministic temporal modulation kernel: a zero-order hold keeps actions constant within each window, filtering high-frequency noise, while a decay profile gradually reduces action magnitude, providing damping against disturbances. As quantified in Figure 6, DWS reduces the AFR by over 80% in precision tasks compared to vanilla baselines (e.g., AFR from 0.56 to 0.10 in Reacher-Hard). Importantly, this improvement preserves responsiveness. Even in highly dynamic environments such as Ball-in-Cup, DWS reduces physical jerk while preserving macroscopic reactivity for successful swing-up and recovery. We also conduct additional experiments on two higher-dimensional DMC locomotion tasks, Cheetah and Walker. While the five low-dimensional tasks provide a controlled testbed for isolating the effect of temporal smoothing, Cheetah and Walker involve more complex body dynamics and larger action spaces, thereby serving as a complementary evaluation of scalability. The complete quantitative results are provided in Appendix D.4. Overall, DWS maintains strong task performance while consistently reducing action fluctuation, jerk, and step-wise action variation. These results further support that the proposed dual-window design provides a robust smooth-control mechanism beyond simple low-dimensional benchmarks.

### 4.3. Case Study of Energy Management Task

We evaluate the proposed DWS on the Fuel Cell Electric Vehicle (FCEV) energy management task within the LearningEMS framework (Wang et al., 2024b), where the objective is to minimize long-term operational cost while maintaining the battery state of charge (SOC) close to the target value of 0.5. Detailed task formulations and experimental settings are deferred to Section E. As reported in Table 1, DWS achieves the best overall performance among all compared methods, obtaining the highest cumulative reward (-1393.15) and the lowest operational cost (157.04). Compared with the strongest baseline, Action

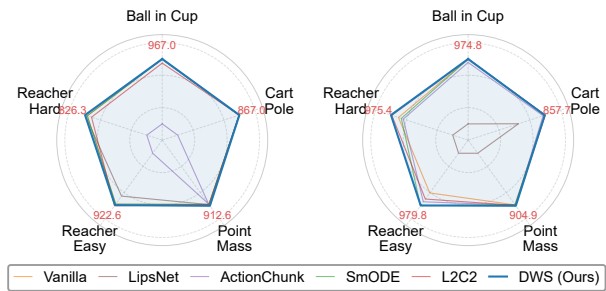

*Figure 4.* **Performance Comparison.** Radar charts showing total returns across five DMC tasks for TD3 (Left) and SAC (Right) backbones. DWS (Blue) achieves the largest coverage area, indicating that it attains SOTA returns without the performance degradation observed in constrained smoothing policies (LipsNet++) or explicit chunking methods (ActionChunk). Full quantitative tables are provided in Section D.

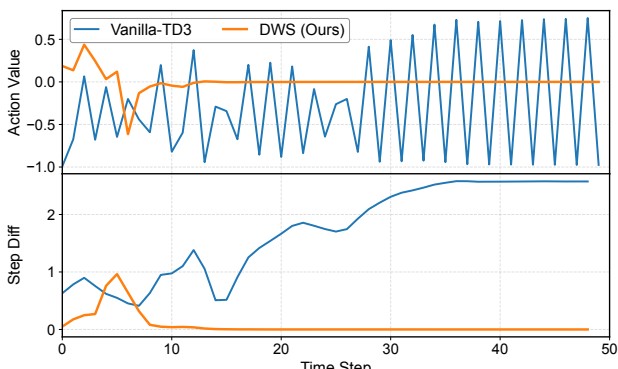

*Figure 5.* **Smoothness Analysis in Reacher-Hard (TD3 Backbone).** Top: Action profile showing that Vanilla-TD3 (Blue) exhibits high-frequency oscillation, while DWS-TD3 (Orange) produces a locally coherent signal. Bottom: The volatility metric ($\|\Delta a_t\|_2$) confirms that DWS improves action smoothness.

Chunk, DWS improves the reward by 56.7% and reduces the operational cost by 10.6%. Although SmODE and L2C2 achieve slightly lower action fluctuation rates (AFR of 0.105 and 0.107, respectively), their aggressive temporal smoothing results in pronounced SOC drift, with final SOC values deviating considerably from the target. In contrast, DWS maintains a more balanced control profile, achieving a moderate AFR of 0.127 while preserving the battery SOC at 0.40. As illustrated in Figure 7, under the WLTC driving cycle, DWS effectively suppresses high-frequency power oscillations during Low-Medium load phases, thereby reducing degradation-related costs, while exhibiting an action-hold behavior under high-load conditions to provide stable high-power output. These behaviors stem directly from the deterministic temporal modulation kernel in the execution window, which enforces temporal coherence and smoothness. Overall, the results demonstrate that DWS balances action smoothness, economic efficiency, and SOC

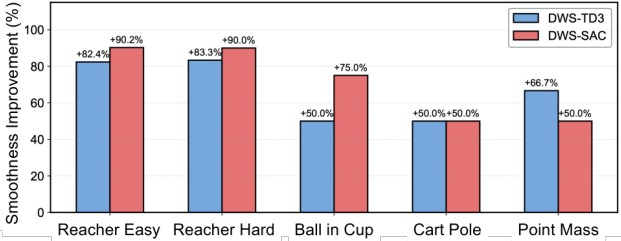

*Figure 6.* **Relative smoothness improvement.** Action smoothness improvements, measured by AFR reduction, of DWS-TD3 and DWS-SAC relative to their Vanilla counterparts across five Deep-Mind benchmarks.

sustainability, making it well suited for long-horizon energy management task. This case study also highlights the key role of action smoothness in real-world control systems.

*Table 1.* **Quantitative comparison results of EMS task.** ↑ higher is better, ↓ lower is better. The target battery SOC is 0.5, closer is better.

| Method | Reward ↑ | Cost (RMB) ↓ | AFR ↓ | SOC |
|---|---|---|---|---|
| Vanilla-TD3 | -10081.97 | 182.39 | 0.205 | 0.22 |
| L2C2 | -7111.91 | 187.37 | 0.107 | 0.26 |
| Action Chunk | -3219.81 | 175.67 | 0.204 | 0.33 |
| SmODE | -16818.18 | 212.69 | **0.105** | 0.17 |
| LipsNet++ | -7723.06 | 232.96 | 0.196 | 0.19 |
| **DWS (Ours)** | **-1393.15** | **157.04** | 0.127 | **0.40** |

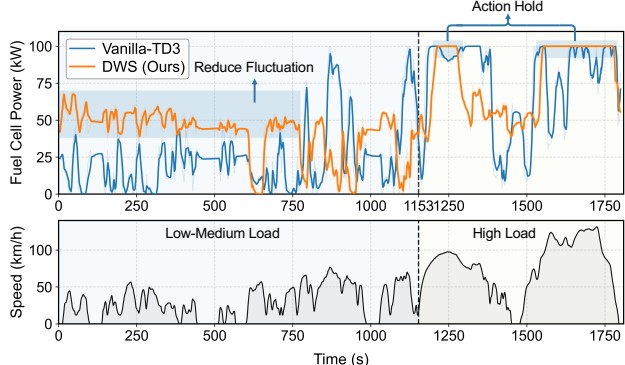

*Figure 7.* **Qualitative analysis of control behaviors under the Light vehicles Test Cycles (WLTC).** DWS exhibits superior smoothness compared to Vanilla-TD3, characterized by reduced fluctuations and stable action holds during high-load.

### 4.4. Case Study of Autonomous Vehicle Task

To evaluate DWS in a high-dimensional, safety-critical domain, we deploy it on end-to-end autonomous driving tasks within the CARLA simulator. In this setting, the agent maps semantic-segmentation observations to task-specific continuous control commands. Crucially, beyond benchmarking control smoothness and safety, these experiments serve to validate DWS's architectural compatibility with EGRL. we

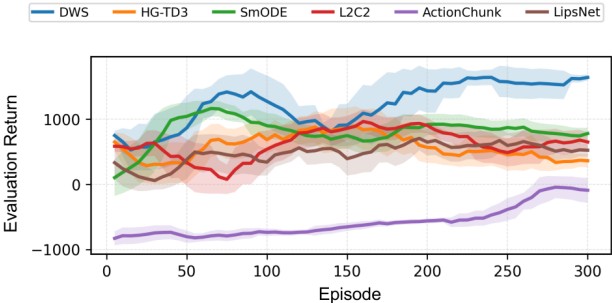

*Figure 8.* **Learning curves on the LCO task.** Evaluation return over training episodes with NPC speed fixed at $5\,\mathrm{m/s}$ (evaluated every five episodes).

adopt human-guided TD3 (HG-TD3) as the backbone algorithm, incorporating human expert intervention strategies during training to accelerate learning and ensure safety. All baselines use the same human-guided mechanism for fair comparison. We test this capability across two distinct scenarios: Lane-Change Overtaking (LCO), which demands precise lateral control, and Autonomous Emergency Braking (AEB), which requires longitudinal stability.

**Main results of LCO.** We first evaluate the LCO task where the ego vehicle must overtake two non-player character (NPC) vehicles (default speed $5\,\mathrm{m/s}$) in a narrow lane without collision or boundary violation. Table 2 demonstrates the dominance of DWS in this lateral control challenge. DWS achieves a 100% success rate with 0% collisions and boundary violations, exhibiting robust compliance with safety constraints. In stark contrast, all baselines struggle significantly; notably, LipsNet++ (2025), a state-of-the-art smooth-policy baseline, only attains a 20% success rate. DWS not only completes the task reliably but does so with superior stability, reducing steering actuation jitter (ActSmooth) by 88.7% and yaw rate oscillation by 78.5% compared to LipsNet++. These gains are further reflected in the learning dynamics ( Figure 8), where DWS converges to higher returns with significantly lower variance than baselines. Robustness analysis (Figure 16) confirms that DWS maintains 100% success across all tested NPC speeds (0-5 $\mathrm{m/s}$), whereas baseline performance deteriorates rapidly as traffic dynamics intensify. Qualitative visualizations ( Figure 9) further show that DWS generates smooth, human-like steering profiles, effectively filtering out the high-frequency corrections typical of standard RL.

**Main results of AEB.** Next, we examine a longitudinal AEB scenario where the agent cruises at $60\,\mathrm{km/h}$ and must brake for a crossing pedestrian. Since all methods achieve 100% success in this simpler setting, the evaluation focuses on the quality of execution—balancing safety (mean time-to-collision on the active braking window, $\mathrm{TTC}_a$) with pas-

*Table 2.* **Experimental results on LCO task (NPC speed 5 m/s).** Task performance and comfort/smoothness metrics. **SR**: Success Rate (↑); **CR**: Collision Rate (↓); **BR**: Beyond-road Rate (↓); **PC**: Path Completion (↑); **ActSmooth**: Action Smoothness (↓); **YawSmooth**: Yaw Angle Smoothness (↓); **YawRate**: Yaw Rate (↓); **AbsAccMean**: Mean Absolute Acceleration (↓); **AbsAccP95**: 95th Percentile of Absolute Acceleration (↓); **Sideslip**$> 3°$: Percentage of time sideslip angle exceeds $3°$ (↓).

| | Task performance (%) | | | | Comfort / smoothness | | | | | |
|---|---|---|---|---|---|---|---|---|---|---|
| Algorithm | SR↑ | CR↓ | BR↓ | PC↑ | ActSmooth↓ | YawSmooth↓ | YawRate↓ | AbsAccMean↓ | AbsAccP95↓ | Sideslip$> 3°$ ↓ |
| HG-TD3 | 0.0 | 75.0 | 25.0 | 68.0 | 0.007±0.002 | 0.002±0.001 | 0.058±0.016 | 0.562±0.095 | 4.084±0.553 | 0.348±0.105 |
| L2C2 | 0.0 | 95.0 | 5.0 | 78.6 | 0.009±0.003 | 0.003±0.001 | 0.068±0.023 | 0.632±0.132 | 4.734±0.887 | 0.428±0.162 |
| ActionChunk | 0.0 | 50.0 | 50.0 | 38.4 | 0.102±0.015 | 0.001±0.000 | 0.030±0.009 | 0.525±0.027 | 4.113±0.171 | 0.315±0.074 |
| SmODE | 0.0 | 100.0 | 0.0 | 87.1 | 0.007±0.002 | 0.003±0.001 | 0.067±0.015 | 0.627±0.097 | 4.504±0.775 | 0.423±0.116 |
| LipsNet++ | 20.0 | 55.0 | 25.0 | 74.3 | 0.012±0.002 | 0.004±0.001 | 0.087±0.020 | 0.833±0.077 | 5.257±0.570 | 0.413±0.087 |
| **DWS (Ours)** | **100.0** | **0.0** | **0.0** | **100.0** | **0.001±0.000** | **0.001±0.000** | **0.019±0.002** | **0.211±0.015** | **1.195±0.144** | **0.071±0.019** |

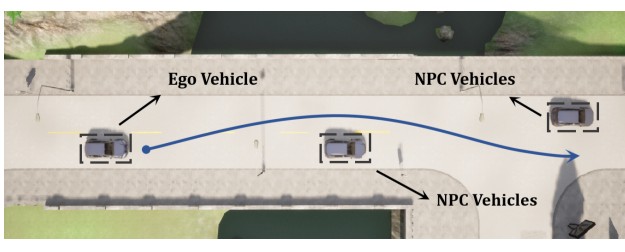

*(a)* Trajectory overview

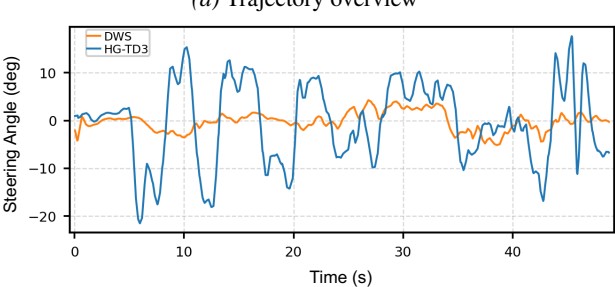

*(b)* Steering angle

*Figure 9.* **Qualitative comparison on the LCO task (example episode).** Trajectory and steering profiles of HG-TD3 (blue) and the proposed DWS (orange).

senger comfort. As detailed in Table 3, DWS achieves the highest reward, effectively expanding the safety margin (increasing $TTC_a$ by 4.9% over HG-TD3) while simultaneously improving comfort (reducing $JerkRMS_a$ by 50.9%). Compared to LipsNet++, DWS yields a +7.5% gain in reward and reduces $AccRMS_a$ by 47.4%. Furthermore, the training return curves ( Figure 19) indicate that DWS attains stable and high returns at the early stage of training. This immediate stability highlights the unique synergy between DWS and expert-guided learning. While baselines often exhibit an unsafe learning phase due to the distributional shift between expert interventions and policy exploration, DWS mitigates this issue. This allows the critic to effectively learn from mixed human-agent trajectories without the temporal misalignment that plagues explicit chunking or standard step-wise methods.

*Table 3.* **AEB Task Results.** Mean±std ($n = 20$). **Reward** (↑): cumulative return. Subscript $a$ denotes metrics computed on the active braking window. $TTC_a$ (↑): mean time-to-collision on the active window. $AccRMS_a$ (↓) and $JerkRMS_a$ (↓): RMS acceleration/jerk on the active window.

| Algorithm | Reward↑ | $TTC_a$↑ | $AccRMS_a$↓ | $JerkRMS_a$↓ |
|---|---|---|---|---|
| HG-TD3 | 362.42±1.06 | 1.708±0.108 | 2.770±0.072 | 97.530±2.570 |
| L2C2 | 361.32±1.64 | 1.677±0.084 | 2.819±0.131 | 99.010±4.530 |
| ActionChunk | 364.58±1.16 | 1.667±0.069 | 1.431±0.083 | 94.270±2.940 |
| SmODE | 368.30±3.38 | 1.327±0.131 | 2.669±0.114 | 93.960±3.980 |
| LipsNet++ | 363.40±0.792 | 1.682±0.052 | 2.633±0.129 | 90.280±4.690 |
| **DWS (Ours)** | **390.70±2.72** | **1.791±0.076** | **1.386±0.128** | **47.900±4.660** |

### 4.5. Execution Profile Comparison

We further compare the two execution profiles used in the execution window on the AEB task: Zero-Order Hold and Dissipative Decay. As reported in Table 4, Zero-Order Hold slightly outperforms Dissipative Decay across reward, safety margin, and comfort metrics. In particular, it achieves higher reward and $TTC_a$, while also yielding lower $AccRMS_a$ and $JerkRMS_a$. These results support our choice of Zero-Order Hold as the default execution profile in the main experiments.

The gap is not large, indicating that both execution profiles can provide effective temporal smoothing once embedded into the DWS framework. However, in this AEB setting, braking is a short-horizon safety-critical response that benefits more from preserving a stable control command over the execution window than from gradually attenuating it. Zero-Order Hold therefore provides a slightly better balance between maintaining braking authority and suppressing unnecessary oscillations, whereas Dissipative Decay introduces mild attenuation that can reduce responsiveness near the critical intervention phase.

This result is also consistent with the role of the execution window in DWS. The main benefit comes from enforcing local temporal coherence under a short commitment horizon, while the exact within-window profile serves as a secondary design choice. In our experiments, Zero-Order Hold is the

*Table 4.* **Comparison of execution profiles on the AEB task.**

| Execution Profile | Reward↑ | TTC$_a$↑ | AccRMS$_a$↓ | JerkRMS$_a$↓ |
|---|---|---|---|---|
| Zero-Order Hold | **390.70**±**2.72** | **1.791**±**0.075** | **1.386**±**0.128** | **47.90**±**4.66** |
| Dissipative Decay | 388.66±4.15 | 1.725±0.103 | 1.413±0.168 | 49.24±5.88 |

*Table 5.* **Sensitivity to the window size** $h$ **across DMC Reacher-Easy and LearningEMS (mean ± std).** For DMC Reacher-Easy, higher reward and lower smoothness are better. For LearningEMS, higher reward and lower cost/AFR are better; SOC is best when closer to the target value 0.5.

| | DMC | | LearningEMS | | | |
|---|---|---|---|---|---|---|
| $h$ | Reward↑ | Smoothness↓ | Reward↑ | Cost↓ | AFR↓ | SOC ($\to 0.5$) |
| 1 | 783.56±143.44 | 0.29±0.32 | -2179±517 | 166.0±6.65 | 0.1497±0.004 | 0.3617±0.0016 |
| 2 | 893.08±168.76 | 0.17±0.29 | -2088±384 | 159.1±6.32 | 0.1335±0.006 | 0.3915±0.0038 |
| 3 | **922.65**±**175.60** | 0.12±0.32 | **-1393**±**437** | **157.0**±**5.70** | 0.1271±0.011 | **0.3956**±**0.027** |
| 4 | 873.76±192.83 | 0.12±0.43 | -1636±422 | 161.2±6.45 | 0.1239±0.014 | 0.3665±0.023 |
| 5 | 845.00±163.35 | **0.11**±**0.29** | -2387±481 | 177.4±7.48 | **0.1189**±**0.021** | 0.3341±0.010 |

*Table 6.* **Sensitivity to the smoothness regularization weight** $\lambda_S$ **across DMC Reacher-Easy and LearningEMS (mean ± std).**

| | DMC | | LearningEMS | | | |
|---|---|---|---|---|---|---|
| $\lambda_S$ | Reward↑ | Smoothness↓ | Reward↑ | Cost↓ | AFR↓ | SOC ($\to 0.5$) |
| 0 | 834.56±177.98 | 0.54±0.21 | -2355±277 | 188.7±4.02 | 0.1807±0.009 | 0.2991±0.041 |
| 0.03 | 879.46±138.43 | 0.21±0.28 | -1951±368 | 181.6±7.50 | 0.1523±0.011 | 0.3373±0.018 |
| 0.10 | **922.65**±**175.60** | 0.12±0.32 | **-1393**±**437** | **157.0**±**5.70** | 0.1271±0.011 | **0.3956**±**0.027** |
| 0.30 | 843.56±163.93 | 0.11±0.43 | -2135±434 | 169.7±5.07 | 0.1183±0.019 | 0.3284±0.035 |
| 0.40 | 804.57±123.78 | **0.09**±**0.45** | -2849±469 | 172.1±4.64 | **0.1037**±**0.002** | 0.2841±0.044 |

more effective default, and Dissipative Decay remains a viable alternative when additional damping is desired.

### 4.6. Sensitivity Analysis

We analyze the sensitivity of DWS to the shared window size $h$ and the smoothness regularization weight $\lambda_S$ on DMC Reacher-Easy and LearningEMS. Tables 5 and 6 show that DWS is reasonably robust across a moderate range of settings. In particular, intermediate settings provide the best overall trade-off between task performance and smoothness. When $h$ or $\lambda_S$ is too small, smoothing is insufficient; when they are too large, control becomes less reactive and task performance degrades despite slightly lower fluctuation metrics. Overall, the default setting $h = 3$ and $\lambda_S = 0.10$ achieves the best balance across both domains.

### 4.7. Ablation Study

We validate the contribution of each DWS component on the main and hardest LCO setting with NPC speed fixed at $5\,\mathrm{m/s}$ ( Table 7). The results again reveal a strong interdependence among components. Each partial variant improves over the HG-TD3 backbone on either task success or motion smoothness, but none matches the complete model. In particular, the execution window alone improves comfort-related metrics but remains limited in task success, while the value window alone substantially boosts success yet still falls short of the full framework. Adding the smoothness

*Table 7.* **Component ablation on the main/hardest LCO setting (NPC speed 5 m/s).** We compare partial variants of DWS under the same evaluation protocol. SR/CR denote success/collision rates, and lower values are better for all smoothness metrics.

| Method | SR↑ | CR↓ | YawSmooth↓ | AbsAccP95↓ | Sideslip$> 3°$ ↓ |
|---|---|---|---|---|---|
| HG-TD3 | 0.00 | 0.75 | 0.0020±0.0010 | 4.084±0.553 | 0.348±0.105 |
| Value Window only | 0.65 | 0.10 | 0.0011±0.0001 | 1.432±0.375 | 0.109±0.044 |
| SmoothReg only | 0.60 | 0.40 | 0.0012±0.0002 | 2.711±0.451 | 0.133±0.044 |
| Value Window + Reg | 0.80 | 0.20 | 0.0008±0.0001 | 1.381±0.080 | 0.079±0.027 |
| Execution Window only | 0.50 | 0.25 | 0.0010±0.0002 | 1.972±1.190 | 0.103±0.085 |
| **DWS (Full)** | **1.00** | **0.00** | **0.0007**±**0.0001** | **1.195**±**0.144** | **0.071**±**0.019** |

regularizer further improves the trade-off, but only the full DWS achieves 100% success with zero collisions while also delivering the best overall smoothness. These results confirm that the three components are complementary, and that their combination is necessary to fully resolve the trade-off between smoothness, safety, and task performance.

## 5. Conclusion

This paper introduces DWS, an implicit action chunking method that balances control smoothness and task performance in continuous RL. By coupling a deterministic execution window with a synchronized value window, DWS enforces temporal coherence and aligns critic learning without increasing the policy action space. An actor-side regularizer further reduces boundary discontinuities. Experiments on diverse benchmarks including the DeepMind benchmarks, industrial energy management, and vision-based autonomous driving show that DWS outperforms SOTA methods in both control smoothness and overall task performance. Compared with explicit action chunking, DWS achieves safer behavior with perfect success rates and reduced jitter. Its design also makes it well-suited for expert-guided RL. This work introduces a new perspective on learning-based control, opening promising avenues for smooth, reliable control in complex systems and supporting real-world applications of RL.

While DWS demonstrates strong empirical performance across diverse domains, its fixed execution horizon introduces a structural inductive bias that may limit policy expressiveness and adaptability in highly dynamic environments, especially when abrupt changes occur within a window. A natural direction for future work is therefore to relax the fixed-schedule assumption through state-dependent or learnable execution profiles, adaptive window lengths, or hybrid schemes that better balance step-wise reactivity with temporally structured execution. Another promising direction is to extend the implicit action chunking framework to offline RL architectures, where its compatibility with standard replay buffers may be particularly advantageous for large-scale pre-training.

## Impact Statement

This paper presents work whose goal is to advance the study of reinforcement learning. There are potential indirect societal consequences of our work, none of which we feel must be specifically highlighted here.

## Acknowledgments

This work was supported by the Research Grants Council of Hong Kong under Grant No. 27206525, and in part by the National Natural Science Foundation of China under Grant No. 62406206 and by the Fundamental Research Funds for the Central Universities.

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

# A. Theoretical Properties of DWS

This appendix formalizes the executed interaction induced by the fixed execution operator $\mathcal{F}_h$ and clarifies the operator view behind the Value Window targets. Throughout, we consider a fixed window length $h$ and discount factor $\gamma \in (0, 1)$.

**Execution operator and executed actions.** At each window boundary $k = mh$ ($m \geq 0$), the policy outputs a reference action $a_k \sim \pi(\cdot \mid s_k)$. The deterministic execution operator $\mathcal{F}_h : \mathcal{A} \to \mathcal{A}^h$ then produces an $h$-step executed profile $\mathbf{u}_{k:k+h-1} = \mathcal{F}_h(a_k)$, which is applied step-by-step to the environment. DWS instantiates $\mathcal{F}_h$ via the modulation profile in Eq. (2).

**Proposition A.1 (Intra-Window Smoothness).** Assume the policy output is bounded such that $\|a_t\| \leq A_{\max}$. For a given execution profile $\mathbf{w}$, the discrete variation of the executed action $u$ within a window is uniformly bounded as

$$\|\Delta u_{t+k}\| \leq |w_k - w_{k-1}| \cdot A_{\max}, \quad \forall k \in \{1, \dots, h-1\}. \tag{10}$$

*Proof.* Within a window starting at time $t$, the execution operator applies the fixed modulation profile

$$u_{t+k} = w_k \, a_t, \qquad k \in \{0, \dots, h-1\}, \tag{11}$$

where $a_t$ is the (cached) reference action sampled at the window boundary. Therefore, for any $k \in \{1, \dots, h-1\}$, the one-step discrete difference satisfies

$$\Delta u_{t+k} := u_{t+k} - u_{t+k-1} = (w_k - w_{k-1}) \, a_t. \tag{12}$$

Taking norms and using the policy bound $\|a_t\| \leq A_{\max}$ yields

$$\|\Delta u_{t+k}\| = \|(w_k - w_{k-1}) \, a_t\| = |w_k - w_{k-1}| \, \|a_t\| \leq |w_k - w_{k-1}| \, A_{\max}, \tag{13}$$

which proves the claim. $\square$

**Proposition A.2 (Augmented Markov view of executed interaction).** Fix $\mathcal{F}_h$ and a policy $\pi$. Define the augmented state as

$$\bar{s}_t := (s_t, \kappa_t, \hat{a}_t), \tag{14}$$

where $\kappa_t \in \{0, \dots, h-1\}$ denotes the within-window phase and $\hat{a}_t$ is the cached reference action for the current window. Then the executed interaction is Markov in $\bar{s}_t$. Consequently, the executed behavior induced by $(\pi, \mathcal{F}_h)$ admits a well-defined primitive action-value function $Q^{\pi^{\mathrm{exec}}}(\bar{s}, u)$.

*Proof sketch.* Given $(\kappa_t, \hat{a}_t)$, the executed action $u_t$ is uniquely determined by the current phase via the fixed profile $\mathcal{F}_h(\hat{a}_t)$, and $\hat{a}_t$ is updated only at window boundaries ($\kappa_t = 0$). Thus, $(s_t, \kappa_t, \hat{a}_t)$ summarizes all controller memory needed to generate future executed actions, so the induced transition dynamics depend only on the current augmented state and executed action.

**Window-aligned $h$-step policy-evaluation operator.** For a fixed executed policy $\pi^{\mathrm{exec}}$ on the augmented process, define the $h$-step policy-evaluation operator $\mathcal{T}_h^{\pi^{\mathrm{exec}}}$ acting on any bounded action-value function $Q$ as

$$\left(\mathcal{T}_h^{\pi^{\mathrm{exec}}} Q\right)(\bar{s}_t, u_t) = \mathbb{E}\left[\sum_{k=0}^{h-1} \gamma^k r_{t+k} + \gamma^h m_t^{(h)} Q(\bar{s}_{t+h}, u_{t+h}) \,\middle|\, \bar{s}_t, u_t\right], \tag{15}$$

where the expectation follows the executed trajectory induced by $(\pi^{\mathrm{exec}}, \mathcal{F}_h)$, and

$$m_t^{(h)} = \mathbf{1}\left[\sum_{k=0}^{h-1} d_{t+k} = 0\right] \tag{16}$$

prevents bootstrapping across termination, matching Eq. (5). For twin critics, the target backup can use $\min(Q_{\theta_1^-}, Q_{\theta_2^-})$ as in the main text.

*Notation.* In the main text we write $Q(s, u)$ for brevity, since $(\kappa_t, \hat{a}_t)$ are internal variables determined by the fixed execution mechanism; formally, the operator acts on $Q(\bar{s}, u)$.

**Proposition A.3 (Operator-Consistent Windowed Target).** Fix $\mathcal{F}_h$ and the induced executed policy $\pi^{\text{exec}}$ on the augmented process. Whenever a valid contiguous executed segment of length $h$ is available (i.e., $z_t = 1$), the windowed target $G_t^{(h)}$ in Eq. (4) is an unbiased empirical sample of the $h$-step executed policy-evaluation backup in Eq. (15) evaluated at $Q_{\theta^-}$:

$$\mathbb{E}\left[ G_t^{(h)} \mid \bar{s}_t, u_t \right] = \left(\mathcal{T}_h^{\pi^{\text{exec}}} Q_{\theta^-}\right)(\bar{s}_t, u_t). \tag{17}$$

Accordingly, the critic objective in Eq. (7) performs Bellman regression toward the window-aligned $h$-step operator when $z_t = 1$, while falling back to one-step TD at episode boundaries or when a full segment is unavailable ($z_t = 0$).

*Proof sketch.* Conditioned on $(\bar{s}_t, u_t)$ and $z_t = 1$, the random return $G_t^{(h)}$ is exactly the Monte Carlo sample of the conditional expectation defining $\left(\mathcal{T}_h^{\pi^{\text{exec}}} Q_{\theta^-}\right)(\bar{s}_t, u_t)$ in Eq. (15), hence is unbiased. $\square$

### A.1. DWS Framework Overview

Figure 10 summarizes how DWS augments a standard off-policy actor–critic loop with a shared horizon $h$. At each window boundary, the actor outputs a reference action $a_t^{\text{RL}}$ and the Execution Window deterministically produces step-wise executed actions $\{u_t, \ldots, u_{t+h-1}\}$. Transitions are stored in replay $\mathcal{D}$ and also appended to the ordered Window Buffer $\mathcal{W}$, from which contiguous length-$h$ segments enable terminal-safe $h$-step targets and a gated critic update; adjacent samples from $\mathcal{W}$ can additionally form the actor-side smoothness regularizer.

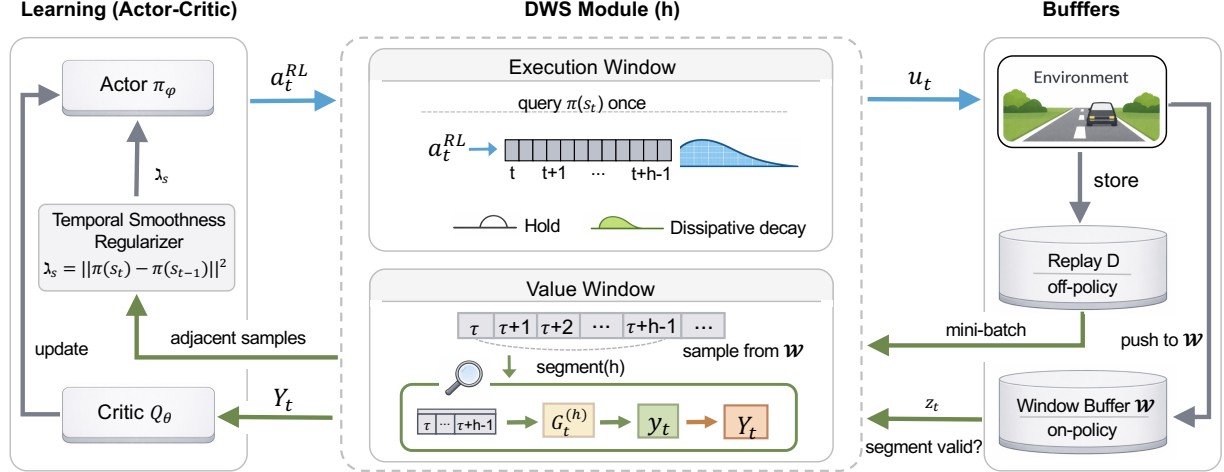

*Figure 10.* **DWS overview (shared horizon $h$).** The Execution Window maps a boundary action $a_t^{\text{RL}}$ to step-wise executed actions $u_t$; replay $\mathcal{D}$ and ordered buffer $\mathcal{W}$ support gated terminal-safe $h$-step critic targets and the actor smoothness regularizer.

### A.2. DWS Pseudocode

---

**Algorithm 1 DWS** (Dual-Window Smoothing)

---

1: **Input:** actor $\pi_\phi$, critic(s) $Q_\theta$, target networks $\phi^-, \theta^-$, replay buffer $\mathcal{D}$, window length $h$, discount $\gamma$, smoothness weight $\lambda_S$
2: Initialize ordered window buffer $\mathcal{W}$ (capacity $\geq h$); execution cache $\mathcal{E} \leftarrow \emptyset$
3: **for** each environment step $t$ **do**
4:     **if window boundary** ($t \bmod h = 0$) **then**
5:         Query reference action $a_t \sim \pi_\phi(\cdot \mid s_t)$
6:         Build executed segment $\mathcal{E}[k] \leftarrow w_k\, a_t$ for $k = 0, \ldots, h-1$       ( Equation (2))
7:     **end if**
8:     Execute $u_t \leftarrow \mathcal{E}[t \bmod h]$ and step environment
9:     Observe $(r_t, s_{t+1}, d_t)$
10:     Store $(s_t, u_t, r_t, s_{t+1}, d_t)$ in replay $\mathcal{D}$
11:     Push $(s_t, u_t, r_t, s_{t+1}, d_t)$ into ordered buffer $\mathcal{W}$
12:     **if** update step **then**
13:         Sample mini-batch from $\mathcal{D}$ and compute one-step targets $y_t$
14:         Sample contiguous length-$h$ segments from $\mathcal{W}$; set $z_t{=}1$ iff the segment exists and does not cross termination
15:         Compute $G_t^{(h)}$ with terminal mask $m_t^{(h)}$       ( Equation (4)–Equation (5))
16:         Form $Y_t = (1 - z_t)y_t + z_t G_t^{(h)}$ and update critic(s)       ( Equation (6)–Equation (7))
17:         Update actor with $L_\pi = L_{\text{base}} + \lambda_S \|\pi_\phi(s_t) - \pi_\phi(s_{t-1})\|_2^2$ using adjacent non-terminal pairs sampled from $\mathcal{W}$ ( Equation (9))
18:         Soft-update target networks $(\phi^-, \theta^-)$
19:     **end if**
20:     $s_t \leftarrow s_{t+1}$
21: **end for**

---

## B. Baselines and ActionChunk instantiation

**Baselines (summary).** Across all experiments, we compare against standard off-policy actor–critic backbones (TD3/SAC or HG-TD3 in CARLA), representative smooth-control methods (L2C2, LipsNet++, SmODE), and an explicit temporal abstraction baseline **ActionChunk**.

**ActionChunk as an explicit action chunking baseline.** **ActionChunk** instantiates the *explicit action chunking* paradigm: at *chunk boundaries* (every $h$ steps), the policy outputs an $h$-step action sequence $\mathbf{a}_{t:t+h-1} = [a_t, \ldots, a_{t+h-1}]$ and executes it open-loop for up to $h$ environment steps. The critic is defined on action sequences $Q(s_t, \mathbf{a}_{t:t+h-1})$ and trained with an $h$-step TD target (with discounting $\gamma^h$, or $\gamma^L$ under early termination within a chunk). We instantiate ActionChunk on top of the backbone used in each domain: TD3 and SAC for DMC, TD3 for EMS, and HG-TD3 for CARLA (denoted ActionChunk (TD3/SAC/HG-TD3) in tables).

**Connection to Q-chunking and scope.** Our ActionChunk implementation follows the core *recipe* described by *Q-chunking* (Li et al., 2025b): (i) chunked policy outputs, (ii) open-loop commitment for $h$ steps, and (iii) a chunked critic enabling an $h$-step backup aligned with temporally extended execution. We do not directly reproduce QC/QC-FQL variants from (Li et al., 2025b), since they incorporate additional behavior-prior modeling and sampling mechanisms (offline-to-online instantiations) that are orthogonal to our goal here: a controlled comparison between *explicit* chunking and our proposed *implicit* chunking (DWS) under a unified online actor–critic pipeline.

## C. Experimental Hyperparameters

We use a unified set of training hyperparameters across all domains to ensure a controlled comparison. Domain-specific settings, including environment configurations, observation and action specifications, and evaluation protocols, are described in their corresponding appendix sections.

*Table 8.* **Shared training hyperparameters used across all experiments.**

| Parameter | Value |
|---|---|
| Replay buffer capacity | 50,000 |
| Batch size | 128 |
| Discount factor $\gamma$ | 0.98 |
| Critic learning rate | $3 \times 10^{-4}$ |
| Actor learning rate | $2 \times 10^{-4}$ |
| Target update coefficient $\tau$ | 0.005 |
| Policy noise | 0.15 |
| Noise clip | 0.5 |
| Policy update frequency | 1 |
| Window horizon $h$ (shared by the Execution/Value Windows) | 3 |
| Smoothness weight $\lambda_S$ | 0.10 |
| Initial exploration scale | 0.5 |
| Minimum exploration scale | 0.005 |
| Exploration decay rate | 0.99988 |

## D. Generalization Analysis on DeepMind Control Suite

While the primary application of **DWS** lies in safety-critical autonomous driving, it is essential to verify that our proposed *Dual-Window* mechanism constitutes a fundamental improvement in continuous control rather than a domain-specific heuristic. To this end, we extend our evaluation to the **DeepMind Control Suite (DMC)** (Tassa et al., 2018). These experiments serve as a rigorous testbed to assess the **generalization capability** of the coupled *Value Window* and *Execution Window* under complex contact dynamics and diverse physical constraints.

### D.1. Environment Rationale and Task Descriptions

We selected four representative tasks that function as surrogate benchmarks for the core challenges addressed by DWS in autonomous driving: precision tracking, dynamic recovery, unstable equilibrium maintenance, and oscillation suppression. Visualizations of these environments are provided in Figure 11.

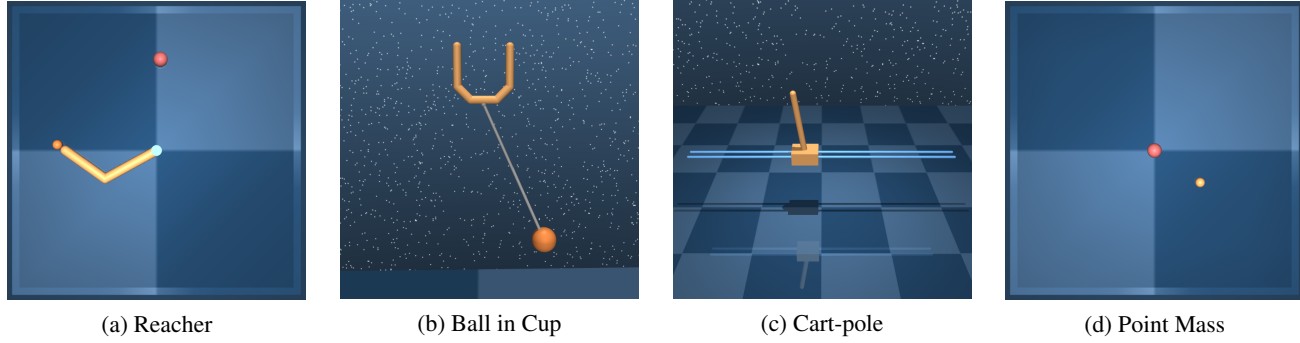

    (a) Reacher          (b) Ball in Cup          (c) Cart-pole          (d) Point Mass

*Figure 11.* **Visualizations of DMC tasks.** These environments act as proxies for core autonomous driving challenges: (a) **Reacher** for precision tracking; (b) **Ball-in-Cup** for impulse handling and dynamic recovery; (c) **Cart-pole** for stabilization of unstable equilibria; and (d) **Point Mass** for inertial control and oscillation suppression.

- **Precision Control (Reacher - Easy & Hard)**: The agent must control a robotic arm to reach a random target with high accuracy. This mirrors **trajectory tracking** and **lane-keeping**. The Reacher-Hard variant, characterized by a minimal target margin, tests the efficacy of the **Value Window**. It verifies whether the critic can correctly assign credit to smooth, fine-grained adjustments required for high-precision tasks.

- **Dynamic Recovery (Ball in Cup - Catch)**: This task requires controlling a cup to catch a ball attached by a string, involving complex impulse dynamics. It serves as a proxy for **recovering from sudden disturbances** (e.g., severe skidding). The task ensures that smoothness constraints do not hinder the agent's ability to execute rapid actions when physically required.

- **Unstable Equilibrium (Cart-pole - Swingup)**: The agent must swing up and balance a pole on a moving cart. This represents a classic underactuated system with a highly unstable equilibrium point, serving as a proxy for **vehicle stability control**. It tests whether DWS maintains responsiveness while smoothing out unnecessary micro-corrections.

- **Inertial Control (Point Mass - Easy)**: This task involves controlling a point mass on a 2D plane to reach a target. In standard RL, policies often exploit high-frequency acceleration changes to maximize instantaneous speed. In our context, Point Mass serves as a proxy for **inertial stability**. We verify whether the **Execution Window** can naturally suppress control jitter (Jerk) and enforce temporally coherent trajectories.

**Experimental Setup:** All environments are simulated using the MuJoCo physics engine (Todorov et al., 2012). We compare our method against several strong baselines, including **Vanilla-TD3**, **Vanilla-SAC**, **Action Chunking** (ActionChunk), and state-of-the-art smoothing methods like **L2C2**, **LipsNet++**, and **SmODE**. DWS is integrated as a plug-and-play module on top of TD3 and SAC backbones. Each algorithm was trained for 1 million time steps across 5 random seeds. ActionChunk is instantiated on both TD3 and SAC backbones in DMC (denoted ActionChunk-TD3 / ActionChunk-SAC); see Section B.

### D.2. Performance and Smoothness Analysis

We present the quantitative results of the DMC generalization experiments in Tables Table 9 through Table 13. We evaluate based on: (1) **Task Performance** (Cumulative Reward), (2) **Action Smoothness** (Smoothness metric and AFR), and (3) **Physical Feasibility** (Jerk and Action Delta).

The advantages of the **Value Window** are most pronounced in the precision-demanding Reacher tasks. In Reacher-Hard (Table 10), where the target margin is minimal, **DWS-SAC** achieves near-optimal performance ($975.35 \pm 13.50$) with negligible variance, significantly outperforming LipsNet++ which collapses in this setting.

In Ball in Cup-Catch (Table 11), standard agents often exhibit high-frequency dithering. **DWS-TD3** achieves the highest reward (966.95) while reducing Jerk-RMS to 0.14. Notably, **ActionChunk-TD3** failed to learn the task, indicating that while Action Chunking captures structure, it struggles with the fluid, low-jerk motions required for dynamic recovery.

For Cart-pole-Swingup (Table 12), **DWS-SAC** achieves a smoothness score of 0.01 and a Jerk-P95 of 0.03, matching the stability of the **L2C2-SAC** baseline but with improved peak rewards (857.73 vs 839.05 for ActionChunk-SAC).

In Point Mass-Easy (Table Table 13), **DWS-SAC** (904.90) outperforms ActionChunk (897.64) and SmODE-SAC (887.15). Crucially, DWS maintains superior smoothness metrics while achieving these high returns, validating its ability to perform efficient inertial control without unnecessary oscillation.

### D.3. Learning Dynamics and Stability

To further analyze the convergence behavior and stability of DWS during training, we visualize the learning curves for all five tasks in Figure 12.

**Reward Convergence (Left Column):** As illustrated in Figure 12e, Figure 12g, Figure 12i, Figure 12a, and Figure 12c, **DWS-SAC** (Red curve) exhibits sample efficiency comparable to or exceeding Vanilla-SAC (Blue curve). In **Impulse tasks** like *Ball in Cup* (Figure 12e), DWS matches the rapid rise of standard RL, proving that the execution window does not delay learning of ballistic motions. In **Precision tasks** like Reacher Hard (Figure 12c), DWS shows superior asymptotic performance. While baselines like L2C2 struggle to stabilize in the narrow target region, DWS maintains a high steady-state reward, attributed to the aligned Value Window preventing the critic from over-valuing jittery corrections.

**Smoothness Evolution (Right Column):** The smoothness profiles (Figure 12f–Figure 12d) reveal the distinct mechanism of DWS. **Rapid Stabilization:** Unlike Vanilla-SAC, which often sees smoothness degrade (increase) as the agent exploits physics exploits, DWS curves drop quickly early in training and remain stable. **Comparison to Baselines:** In Point Mass (Figure 12j), DWS maintains the lowest smoothness metric throughout training compared to SmODE and ActionChunk, indicating that the generated trajectory is fundamentally more coherent and less reliant on high-frequency actuation.

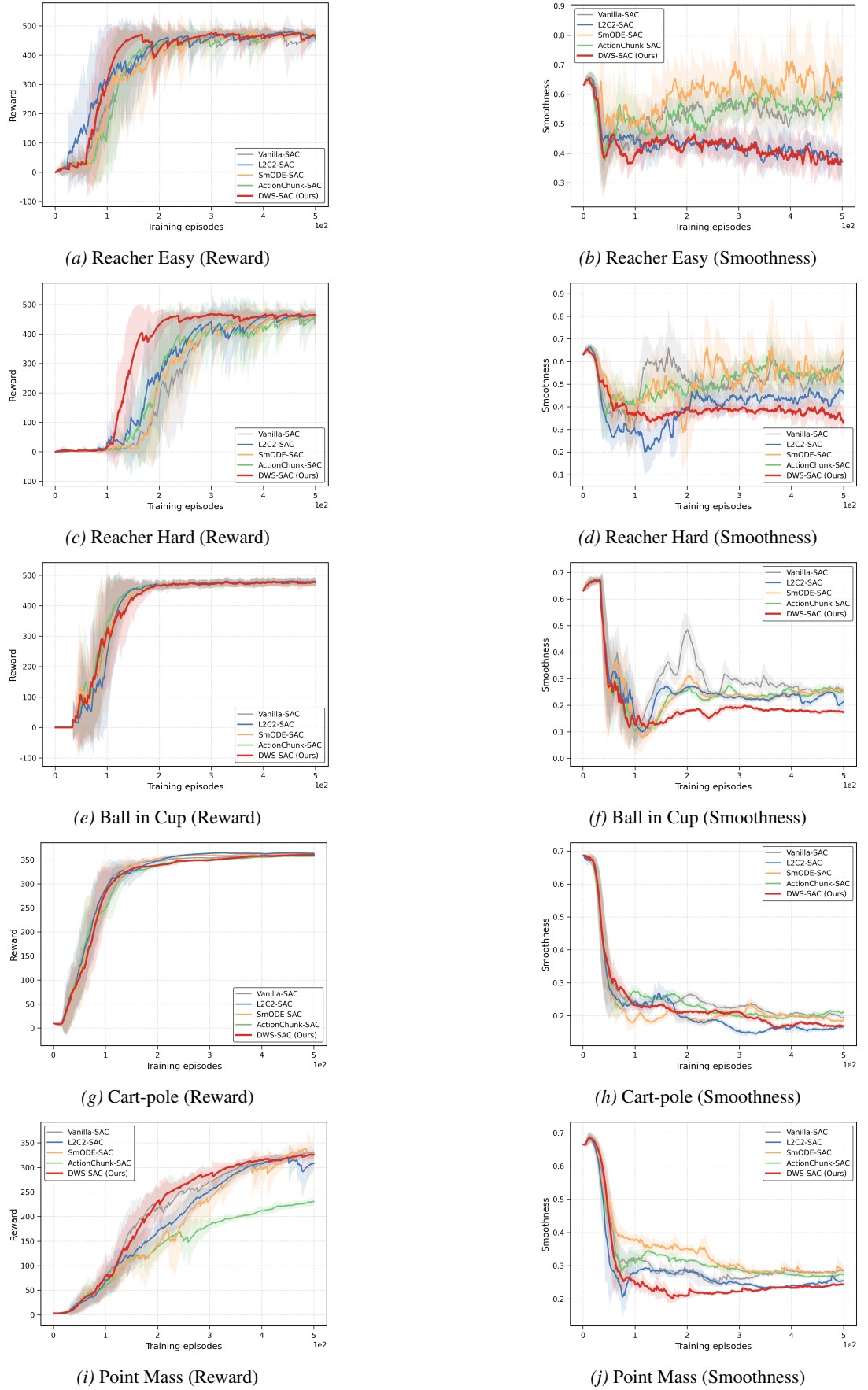

*(a)* Reacher Easy (Reward)

*(b)* Reacher Easy (Smoothness)

*(c)* Reacher Hard (Reward)

*(d)* Reacher Hard (Smoothness)

*(e)* Ball in Cup (Reward)

*(f)* Ball in Cup (Smoothness)

*(g)* Cart-pole (Reward)

*(h)* Cart-pole (Smoothness)

*(i)* Point Mass (Reward)

*(j)* Point Mass (Smoothness)

*Table 9.* Quantitative results on **Reacher - Easy**. We report the mean $\pm$ standard deviation over 5 seeds. $\uparrow$ indicates higher is better; $\downarrow$ indicates lower is better.

| Algorithm | Reward $\uparrow$ | Smooth. $\downarrow$ | AFR_L2 $\downarrow$ | AFR_L1 $\downarrow$ | Jerk_RMS $\downarrow$ | Jerk_P95 $\downarrow$ | Delta_Max $\downarrow$ | Delta_P95 $\downarrow$ |
|---|---|---|---|---|---|---|---|---|
| Vanilla-TD3 | $908.20 \pm 216.72$ | $0.68 \pm 0.57$ | $1.03 \pm 0.84$ | $1.35 \pm 1.14$ | $2.15 \pm 1.64$ | $2.50 \pm 1.73$ | $1.91 \pm 0.54$ | $1.29 \pm 0.87$ |
| L2C2-TD3 | $911.35 \pm 207.40$ | $0.57 \pm 0.52$ | $0.84 \pm 0.75$ | $1.15 \pm 1.05$ | $1.76 \pm 1.47$ | $1.82 \pm 1.51$ | $1.58 \pm 0.40$ | $0.93 \pm 0.74$ |
| LipsNet++-TD3 | $794.50 \pm 387.64$ | $0.54 \pm 0.70$ | $0.81 \pm 1.00$ | $1.09 \pm 1.39$ | $1.67 \pm 1.99$ | $1.70 \pm 2.03$ | $1.46 \pm 0.73$ | $0.88 \pm 1.01$ |
| SmODE-TD3 | $914.22 \pm 198.62$ | $0.54 \pm 0.53$ | $1.01 \pm 0.63$ | $1.20 \pm 1.08$ | $1.98 \pm 1.34$ | $2.03 \pm 1.63$ | $1.71 \pm 0.64$ | $1.08 \pm 0.79$ |
| ActionChunk-TD3 | $74.60 \pm 101.01$ | - | - | - | - | - | - | - |
| DWS-TD3 (Ours) | $\mathbf{922.65 \pm 175.60}$ | $\mathbf{0.12 \pm 0.32}$ | $\mathbf{0.19 \pm 0.49}$ | $\mathbf{0.24 \pm 0.63}$ | $\mathbf{0.57 \pm 0.93}$ | $\mathbf{0.64 \pm 1.41}$ | $\mathbf{1.56 \pm 0.47}$ | $\mathbf{0.33 \pm 0.71}$ |
| Vanilla-SAC | $792.05 \pm 386.56$ | $0.41 \pm 0.88$ | $0.65 \pm 1.39$ | $0.83 \pm 1.76$ | $1.40 \pm 2.76$ | $1.34 \pm 2.84$ | $1.90 \pm 1.01$ | $0.69 \pm 1.41$ |
| L2C2-SAC | $883.60 \pm 301.64$ | $0.03 \pm 0.06$ | $\mathbf{0.05 \pm 0.09}$ | $\mathbf{0.06 \pm 0.12}$ | $\mathbf{0.08 \pm 0.08}$ | $\mathbf{0.08 \pm 0.17}$ | $0.84 \pm 0.29$ | $0.09 \pm 0.19$ |
| LipsNet++-SAC | $57.20 \pm 190.32$ | - | - | - | - | - | - | - |
| SmODE-SAC | $984.65 \pm 14.83$ | $0.40 \pm 0.50$ | $0.62 \pm 0.73$ | $0.80 \pm 1.00$ | $1.29 \pm 1.39$ | $1.32 \pm 1.50$ | $1.82 \pm 0.54$ | $0.75 \pm 0.80$ |
| ActionChunk-SAC | $924.95 \pm 22.97$ | $0.03 \pm 0.08$ | $0.05 \pm 0.13$ | $0.06 \pm 0.16$ | $0.12 \pm 0.16$ | $0.10 \pm 0.33$ | $0.89 \pm 0.34$ | $0.11 \pm 0.32$ |
| DWS-SAC (Ours) | $\mathbf{979.75 \pm 19.50}$ | $\mathbf{0.04 \pm 0.18}$ | $0.08 \pm 0.32$ | $0.09 \pm 0.35$ | $0.18 \pm 0.64$ | $0.16 \pm 0.64$ | $\mathbf{0.70 \pm 0.34}$ | $\mathbf{0.09 \pm 0.32}$ |

*Table 10.* Quantitative results on **Reacher - Hard**. We report the mean $\pm$ standard deviation over 5 seeds. $\uparrow$ indicates higher is better; $\downarrow$ indicates lower is better.

| Algorithm | Reward $\uparrow$ | Smooth. $\downarrow$ | AFR_L2 $\downarrow$ | AFR_L1 $\downarrow$ | Jerk_RMS $\downarrow$ | Jerk_P95 $\downarrow$ | Delta_Max $\downarrow$ | Delta_P95 $\downarrow$ |
|---|---|---|---|---|---|---|---|---|
| Vanilla-TD3 | $802.00 \pm 350.85$ | $0.36 \pm 0.44$ | $0.56 \pm 0.65$ | $0.73 \pm 0.87$ | $1.23 \pm 1.19$ | $1.58 \pm 1.45$ | $1.81 \pm 0.49$ | $0.89 \pm 0.84$ |
| L2C2-TD3 | $757.05 \pm 384.13$ | $\mathbf{0.04 \pm 0.09}$ | $\mathbf{0.07 \pm 0.14}$ | $\mathbf{0.08 \pm 0.17}$ | $0.30 \pm 0.26$ | $0.23 \pm 0.47$ | $1.37 \pm 0.47$ | $0.14 \pm 0.25$ |
| LipsNet++-TD3 | $824.00 \pm 351.69$ | $0.63 \pm 0.47$ | $0.95 \pm 0.69$ | $1.26 \pm 0.95$ | $1.92 \pm 1.37$ | $2.09 \pm 1.44$ | $1.85 \pm 0.48$ | $1.10 \pm 0.76$ |
| SmODE-TD3 | $802.00 \pm 350.85$ | $0.36 \pm 0.44$ | $0.56 \pm 0.65$ | $0.73 \pm 0.87$ | $1.23 \pm 1.19$ | $1.58 \pm 1.45$ | $1.81 \pm 0.49$ | $0.89 \pm 0.84$ |
| ActionChunk-TD3 | $4.80 \pm 13.59$ | - | - | - | - | - | - | - |
| DWS-TD3 (Ours) | $\mathbf{826.30 \pm 353.78}$ | $0.06 \pm 0.14$ | $0.10 \pm 0.24$ | $0.12 \pm 0.28$ | $\mathbf{0.27 \pm 0.48}$ | $\mathbf{0.23 \pm 0.58}$ | $\mathbf{0.98 \pm 0.31}$ | $\mathbf{0.12 \pm 0.29}$ |
| Vanilla-SAC | $880.55 \pm 301.64$ | $0.10 \pm 0.35$ | $0.15 \pm 0.51$ | $0.20 \pm 0.70$ | $0.43 \pm 1.05$ | $0.39 \pm 1.16$ | $1.84 \pm 0.51$ | $0.21 \pm 0.58$ |
| L2C2-SAC | $964.30 \pm 20.71$ | $0.01 \pm 0.04$ | $0.02 \pm 0.07$ | $0.03 \pm 0.08$ | $0.07 \pm 0.07$ | $0.04 \pm 0.14$ | $0.99 \pm 0.32$ | $0.04 \pm 0.13$ |
| LipsNet++-SAC | $0.40 \pm 1.05$ | - | - | - | - | - | - | - |
| SmODE-SAC | $844.15 \pm 322.87$ | $0.44 \pm 0.42$ | $0.68 \pm 0.62$ | $0.88 \pm 0.83$ | $1.41 \pm 1.22$ | $1.78 \pm 1.42$ | $1.66 \pm 0.55$ | $0.97 \pm 0.77$ |
| ActionChunk-SAC | $819.15 \pm 356.27$ | $0.20 \pm 0.39$ | $0.29 \pm 0.56$ | $0.39 \pm 0.77$ | $0.61 \pm 1.11$ | $0.61 \pm 1.16$ | $1.10 \pm 0.44$ | $0.34 \pm 0.58$ |
| DWS-SAC (Ours) | $\mathbf{975.35 \pm 13.50}$ | $\mathbf{0.00 \pm 0.00}$ | $\mathbf{0.01 \pm 0.00}$ | $\mathbf{0.01 \pm 0.00}$ | $\mathbf{0.04 \pm 0.02}$ | $\mathbf{0.00 \pm 0.01}$ | $\mathbf{0.78 \pm 0.25}$ | $\mathbf{0.00 \pm 0.01}$ |

## D.4. Additional Results on High-Dimensional DMC Locomotion Tasks

To further examine the scalability of DWS in more complex continuous control scenarios, we conduct additional experiments on two high-dimensional DMC locomotion tasks, Cheetah and Walker. Compared with the lower-dimensional tasks reported in the main text, Cheetah and Walker involve more complex body dynamics, higher-dimensional action spaces, and stronger temporal coupling, making them useful benchmarks for evaluating both control smoothness and policy stability in high-dimensional locomotion. Visualizations of the two high-dimensional locomotion environments are provided in Figure 13.

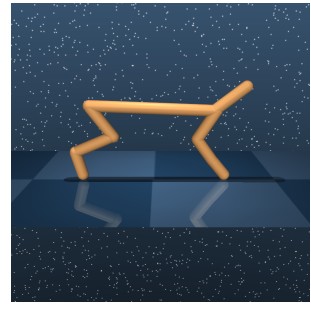

(a) Cheetah        (b) Walker

*Figure 13.* **Visualizations of high-dimensional DMC locomotion tasks.** We further evaluate DWS on two challenging locomotion benchmarks: (a) **Cheetah**, which requires fast and coordinated limb control for forward running; and (b) **Walker**, which requires stable full-body balance and smooth gait generation. These tasks involve more complex body dynamics and higher-dimensional action spaces than the low-dimensional DMC tasks, providing a complementary evaluation of DWS in dynamic locomotion control.

*Table 11.* Quantitative results on **Ball in Cup - Catch**. We report the mean $\pm$ standard deviation over 5 seeds. $\uparrow$ indicates higher is better; $\downarrow$ indicates lower is better.

| Algorithm | Reward $\uparrow$ | Smooth. $\downarrow$ | AFR_L2 $\downarrow$ | AFR_L1 $\downarrow$ | Jerk_RMS $\downarrow$ | Jerk_P95 $\downarrow$ | Delta_Max $\downarrow$ | Delta_P95 $\downarrow$ |
|---|---|---|---|---|---|---|---|---|
| Vanilla-TD3 | $966.40 \pm 8.93$ | $0.02 \pm 0.01$ | $0.03 \pm 0.01$ | $0.03 \pm 0.01$ | $0.30 \pm 0.11$ | $0.02 \pm 0.02$ | $2.12 \pm 0.50$ | $0.01 \pm 0.01$ |
| L2C2-TD3 | $917.20 \pm 216.37$ | $0.07 \pm 0.02$ | $0.15 \pm 0.03$ | $0.15 \pm 0.03$ | $0.32 \pm 0.07$ | $0.48 \pm 0.11$ | $\mathbf{1.44 \pm 0.65}$ | $0.26 \pm 0.06$ |
| LipsNet++-TD3 | $963.40 \pm 21.77$ | $0.01 \pm 0.01$ | $0.02 \pm 0.02$ | $0.02 \pm 0.02$ | $0.21 \pm 0.16$ | $0.06 \pm 0.12$ | $1.66 \pm 0.93$ | $0.03 \pm 0.05$ |
| SmODE-TD3 | $965.40 \pm 8.93$ | $0.02 \pm 0.01$ | $0.03 \pm 0.01$ | $0.03 \pm 0.01$ | $0.30 \pm 0.11$ | $0.02 \pm 0.02$ | $2.12 \pm 0.50$ | $0.01 \pm 0.01$ |
| ActionChunk-TD3 | $3.12 \pm 0.08$ | - | - | - | - | - | - | - |
| DWS-TD3 (Ours) | $\mathbf{966.95 \pm 9.17}$ | $\mathbf{0.01 \pm 0.00}$ | $\mathbf{0.01 \pm 0.00}$ | $\mathbf{0.01 \pm 0.01}$ | $\mathbf{0.14 \pm 0.05}$ | $\mathbf{0.00 \pm 0.00}$ | $2.34 \pm 0.81$ | $\mathbf{0.00 \pm 0.00}$ |
| Vanilla-SAC | $974.25 \pm 19.65$ | $0.04 \pm 0.03$ | $0.07 \pm 0.04$ | $0.09 \pm 0.06$ | $0.48 \pm 0.22$ | $0.40 \pm 0.43$ | $5.17 \pm 1.84$ | $0.26 \pm 0.30$ |
| L2C2-SAC | $972.50 \pm 14.55$ | $0.01 \pm 0.00$ | $0.01 \pm 0.01$ | $0.01 \pm 0.01$ | $0.11 \pm 0.06$ | $\mathbf{0.01 \pm 0.01}$ | $1.31 \pm 0.56$ | $\mathbf{0.01 \pm 0.01}$ |
| LipsNet++-SAC | $49.65 \pm 222.04$ | - | - | - | - | - | - | - |
| SmODE-SAC | $972.65 \pm 12.55$ | $0.01 \pm 0.00$ | $0.02 \pm 0.01$ | $0.02 \pm 0.01$ | $0.22 \pm 0.06$ | $0.05 \pm 0.07$ | $2.05 \pm 0.36$ | $0.03 \pm 0.04$ |
| ActionChunk-SAC | $933.20 \pm 13.51$ | $0.28 \pm 0.05$ | $0.41 \pm 0.07$ | $0.56 \pm 0.09$ | $0.88 \pm 0.07$ | $1.04 \pm 0.01$ | $1.87 \pm 0.36$ | $0.53 \pm 0.00$ |
| DWS-SAC (Ours) | $\mathbf{974.75 \pm 11.87}$ | $\mathbf{0.01 \pm 0.00}$ | $\mathbf{0.01 \pm 0.01}$ | $\mathbf{0.01 \pm 0.01}$ | $\mathbf{0.11 \pm 0.05}$ | $0.02 \pm 0.02$ | $\mathbf{1.16 \pm 0.54}$ | $0.02 \pm 0.02$ |

*Table 12.* Quantitative results on **Cart Pole - Swingup**. We report the mean $\pm$ standard deviation over 5 seeds. $\uparrow$ indicates higher is better; $\downarrow$ indicates lower is better.

| Algorithm | Reward $\uparrow$ | Smooth. $\downarrow$ | AFR_L2 $\downarrow$ | AFR_L1 $\downarrow$ | Jerk_RMS $\downarrow$ | Jerk_P95 $\downarrow$ | Delta_Max $\downarrow$ | Delta_P95 $\downarrow$ |
|---|---|---|---|---|---|---|---|---|
| Vanilla-TD3 | $863.63 \pm 0.28$ | $0.01 \pm 0.00$ | $0.01 \pm 0.00$ | $0.01 \pm 0.00$ | $0.03 \pm 0.00$ | $0.03 \pm 0.00$ | $\mathbf{0.48 \pm 0.05}$ | $0.04 \pm 0.00$ |
| L2C2-TD3 | $864.58 \pm 0.20$ | $0.01 \pm 0.00$ | $0.01 \pm 0.00$ | $0.01 \pm 0.00$ | $0.03 \pm 0.00$ | $0.04 \pm 0.00$ | $\mathbf{0.48 \pm 0.05}$ | $0.06 \pm 0.00$ |
| LipsNet++-TD3 | $864.12 \pm 0.30$ | $0.01 \pm 0.00$ | $0.01 \pm 0.00$ | $0.01 \pm 0.00$ | $0.04 \pm 0.01$ | $0.04 \pm 0.00$ | $0.65 \pm 0.15$ | $0.05 \pm 0.00$ |
| SmODE-TD3 | $863.63 \pm 0.28$ | $0.01 \pm 0.00$ | $0.01 \pm 0.00$ | $0.01 \pm 0.00$ | $0.03 \pm 0.00$ | $0.03 \pm 0.00$ | $\mathbf{0.48 \pm 0.05}$ | $0.04 \pm 0.00$ |
| ActionChunk-TD3 | $0.01 \pm 0.01$ | - | - | - | - | - | - | - |
| DWS-TD3 (Ours) | $\mathbf{867.00 \pm 0.20}$ | $\mathbf{0.01 \pm 0.00}$ | $\mathbf{0.01 \pm 0.00}$ | $\mathbf{0.01 \pm 0.00}$ | $\mathbf{0.03 \pm 0.00}$ | $\mathbf{0.03 \pm 0.00}$ | $0.54 \pm 0.03$ | $\mathbf{0.04 \pm 0.00}$ |
| Vanilla-SAC | $852.85 \pm 0.19$ | $0.02 \pm 0.00$ | $0.02 \pm 0.00$ | $0.02 \pm 0.00$ | $0.14 \pm 0.01$ | $0.10 \pm 0.01$ | $1.73 \pm 0.11$ | $0.13 \pm 0.01$ |
| L2C2-SAC | $\mathbf{863.94 \pm 0.35}$ | $0.01 \pm 0.00$ | $0.01 \pm 0.00$ | $0.01 \pm 0.00$ | $0.02 \pm 0.00$ | $0.04 \pm 0.00$ | $0.31 \pm 0.02$ | $0.07 \pm 0.00$ |
| LipsNet++-SAC | $563.47 \pm 9.29$ | - | - | - | - | - | - | - |
| SmODE-SAC | $861.80 \pm 0.28$ | $0.02 \pm 0.00$ | $0.02 \pm 0.00$ | $0.02 \pm 0.00$ | $0.13 \pm 0.01$ | $0.09 \pm 0.01$ | $1.73 \pm 0.17$ | $0.10 \pm 0.01$ |
| ActionChunk-SAC | $839.05 \pm 0.26$ | $0.01 \pm 0.00$ | $0.01 \pm 0.00$ | $0.01 \pm 0.00$ | $0.07 \pm 0.01$ | $0.05 \pm 0.01$ | $0.91 \pm 0.13$ | $0.06 \pm 0.00$ |
| DWS-SAC (Ours) | $857.73 \pm 0.27$ | $\mathbf{0.01 \pm 0.00}$ | $\mathbf{0.01 \pm 0.00}$ | $\mathbf{0.01 \pm 0.00}$ | $\mathbf{0.02 \pm 0.00}$ | $\mathbf{0.03 \pm 0.00}$ | $\mathbf{0.31 \pm 0.01}$ | $\mathbf{0.07 \pm 0.00}$ |

The results are summarized in Tables 14 and 15. Overall, DWS demonstrates a stronger balance between task performance and control smoothness on both tasks. On Cheetah, DWS achieves the highest reward under both TD3 and SAC backbones, while obtaining the best or near-best results on most smoothness-related metrics. In particular, with the SAC backbone, DWS achieves the best performance across all reported smoothness metrics, indicating that it can significantly reduce action fluctuation, jerk, and step-wise action variation while maintaining high task return.

On Walker, DWS also shows consistent advantages. It is worth noting that, under the TD3 backbone, LipsNet++ obtains a slightly higher average reward than DWS-TD3. However, DWS-TD3 achieves the best results across all smoothness-related metrics while maintaining a highly competitive return close to the best-performing baseline. Therefore, from the perspective of the overall trade-off between task performance and smooth control, DWS remains the most favorable choice. Under the SAC backbone, DWS achieves both the highest reward and the best smoothness performance. These results suggest that the proposed dual-window design is not limited to low-dimensional control tasks, but can also scale to more complex and dynamic high-dimensional locomotion scenarios, where smooth, stable, and responsive control is simultaneously required.

## E. Additional Experiments on Energy Management Task

We evaluate our proposed algorithm on the Energy Management Strategy (EMS) task using **LearningEMS**, a high-fidelity unified framework for electric vehicle (EV) control. We focus specifically on the Fuel Cell Electric Vehicle (FCEV) benchmark, which stresses the agent's ability to optimize long-horizon energy consumption while strictly adhering to physical constraints (e.g., battery State of Charge) and managing the degradation of the fuel cell stack(Li et al., 2024; Wang et al., 2023).

*Figure 12.* Learning curves on DeepMind Control Suite benchmarks. **Left:** Evaluation return over 1M steps (higher is better). **Right:** Smoothness metric over training (lower is better). DWS (Red) is compared against standard baselines (TD3/SAC) and smoothing methods (L2C2, SmODE, ActionChunk). Shaded regions represent standard deviation ($n = 5$).

*Table 13.* Quantitative results on **Point Mass - Easy**. We report the mean $\pm$ standard deviation over 5 seeds. $\uparrow$ indicates higher is better; $\downarrow$ indicates lower is better.

| Algorithm | Reward $\uparrow$ | Smooth. $\downarrow$ | AFR_L2 $\downarrow$ | AFR_L1 $\downarrow$ | Jerk_RMS $\downarrow$ | Jerk_P95 $\downarrow$ | Delta_Max $\downarrow$ | Delta_P95 $\downarrow$ |
|---|---|---|---|---|---|---|---|---|
| Vanilla-TD3 | $896.09 \pm 33.48$ | $0.03 \pm 0.10$ | $0.01 \pm 0.00$ | $0.02 \pm 0.00$ | $0.01 \pm 0.00$ | $0.01 \pm 0.01$ | $0.17 \pm 0.06$ | $0.32 \pm 0.08$ |
| L2C2-TD3 | $894.52 \pm 32.40$ | $0.03 \pm 0.02$ | $0.03 \pm 0.02$ | $0.01 \pm 0.00$ | $0.01 \pm 0.00$ | $0.01 \pm 0.00$ | $0.18 \pm 0.09$ | $0.22 \pm 0.02$ |
| LipsNet++-TD3 | $896.48 \pm 28.86$ | $0.02 \pm 0.01$ | $0.01 \pm 0.01$ | $0.01 \pm 0.01$ | $0.01 \pm 0.01$ | $0.01 \pm 0.00$ | $0.28 \pm 0.11$ | $0.22 \pm 0.11$ |
| SmODE-TD3 | $892.73 \pm 21.56$ | $0.02 \pm 0.01$ | $0.02 \pm 0.01$ | $0.01 \pm 0.01$ | $0.02 \pm 0.01$ | $0.03 \pm 0.01$ | $0.14 \pm 0.03$ | $0.19 \pm 0.04$ |
| ActionChunk-TD3 | $886.45 \pm 22.98$ | $0.02 \pm 0.02$ | $0.03 \pm 0.01$ | $0.02 \pm 0.01$ | $0.03 \pm 0.01$ | $0.02 \pm 0.01$ | $0.28 \pm 0.10$ | $0.26 \pm 0.12$ |
| DWS-TD3 (Ours) | $\mathbf{912.65 \pm 29.83}$ | $\mathbf{0.01 \pm 0.01}$ | $\mathbf{0.01 \pm 0.01}$ | $\mathbf{0.01 \pm 0.01}$ | $\mathbf{0.01 \pm 0.00}$ | $\mathbf{0.01 \pm 0.01}$ | $\mathbf{0.05 \pm 0.01}$ | $\mathbf{0.15 \pm 0.07}$ |
| Vanilla-SAC | $893.25 \pm 24.84$ | $0.02 \pm 0.01$ | $0.02 \pm 0.01$ | $0.02 \pm 0.01$ | $0.04 \pm 0.02$ | $0.03 \pm 0.01$ | $0.19 \pm 0.06$ | $0.15 \pm 0.02$ |
| L2C2-SAC | $898.71 \pm 13.64$ | $0.03 \pm 0.02$ | $0.02 \pm 0.01$ | $0.02 \pm 0.01$ | $0.02 \pm 0.01$ | $0.03 \pm 0.02$ | $0.12 \pm 0.08$ | $0.13 \pm 0.04$ |
| LipsNet++-SAC | $24.94 \pm 15.83$ | - | - | - | - | - | - | - |
| SmODE-SAC | $887.15 \pm 16.55$ | $0.01 \pm 0.00$ | $0.03 \pm 0.01$ | $0.02 \pm 0.01$ | $0.03 \pm 0.02$ | $0.03 \pm 0.02$ | $0.12 \pm 0.06$ | $0.41 \pm 0.04$ |
| ActionChunk-SAC | $897.64 \pm 14.91$ | $0.02 \pm 0.01$ | $0.04 \pm 0.01$ | $0.05 \pm 0.02$ | $1.11 \pm 0.02$ | $0.06 \pm 0.07$ | $0.21 \pm 0.10$ | $0.33 \pm 0.04$ |
| DWS-SAC (Ours) | $\mathbf{904.90 \pm 30.11}$ | $\mathbf{0.01 \pm 0.00}$ | $0.02 \pm 0.01$ | $\mathbf{0.01 \pm 0.00}$ | $\mathbf{0.01 \pm 0.01}$ | $0.02 \pm 0.01$ | $0.10 \pm 0.03$ | $\mathbf{0.11 \pm 0.02}$ |

*Table 14.* **Performance comparison on the harder DMC Cheetah task.** Mean$\pm$std over 5 seeds. Higher reward is better; all smoothness-related metrics are lower-is-better. Best results within each backbone block (TD3 or SAC) are highlighted in **bold**.

| Algorithm | Reward$\uparrow$ | Smooth.$\downarrow$ | AFR$_{L2}\downarrow$ | AFR$_{L1}\downarrow$ | Jerk$_{RMS}\downarrow$ | Jerk$_{P95}\downarrow$ | Delta$_{max}\downarrow$ | Delta$_{P95}\downarrow$ |
|---|---|---|---|---|---|---|---|---|
| Vanilla-TD3 | $456.77 \pm 153.82$ | $0.21 \pm 0.02$ | $0.89 \pm 0.07$ | $1.49 \pm 0.12$ | $1.64 \pm 0.13$ | $3.22 \pm 0.26$ | $3.68 \pm 0.31$ | $2.38 \pm 0.09$ |
| L2C2-TD3 | $402.06 \pm 152.20$ | $0.18 \pm 0.03$ | $0.78 \pm 0.14$ | $1.03 \pm 0.18$ | $1.53 \pm 0.15$ | $2.74 \pm 0.26$ | $\mathbf{2.70 \pm 0.32}$ | $2.48 \pm 0.20$ |
| LipsNet++-TD3 | $591.46 \pm 81.29$ | $0.24 \pm 0.00$ | $0.97 \pm 0.02$ | $1.45 \pm 0.03$ | $1.76 \pm 0.03$ | $3.56 \pm 0.15$ | $4.07 \pm 0.13$ | $2.35 \pm 0.07$ |
| SmODE-TD3 | $434.74 \pm 169.08$ | $0.23 \pm 0.02$ | $0.97 \pm 0.08$ | $1.37 \pm 0.14$ | $1.74 \pm 0.13$ | $3.22 \pm 0.26$ | $3.68 \pm 0.31$ | $2.38 \pm 0.09$ |
| ActionChunk-TD3 | $0.01 \pm 0.01$ | - | - | - | - | - | - | - |
| DWS-TD3 (Ours) | $\mathbf{595.35 \pm 74.63}$ | $\mathbf{0.16 \pm 0.03}$ | $\mathbf{0.68 \pm 0.11}$ | $\mathbf{0.96 \pm 0.16}$ | $\mathbf{1.22 \pm 0.17}$ | $\mathbf{2.50 \pm 0.25}$ | $3.45 \pm 0.25$ | $\mathbf{1.91 \pm 0.17}$ |
| Vanilla-SAC | $777.35 \pm 24.84$ | $0.41 \pm 0.01$ | $1.24 \pm 0.03$ | $2.45 \pm 0.05$ | $1.74 \pm 0.08$ | $3.38 \pm 0.19$ | $7.29 \pm 2.16$ | $2.75 \pm 0.12$ |
| L2C2-SAC | $432.90 \pm 183.74$ | $0.13 \pm 0.04$ | $0.50 \pm 0.17$ | $0.78 \pm 0.26$ | $0.78 \pm 0.15$ | $1.63 \pm 0.22$ | $3.12 \pm 0.28$ | $1.33 \pm 0.14$ |
| LipsNet++-SAC | $347.98 \pm 15.83$ | $0.40 \pm 0.00$ | $1.20 \pm 0.01$ | $2.43 \pm 0.02$ | $1.31 \pm 0.04$ | $2.89 \pm 0.18$ | $4.71 \pm 0.49$ | $2.38 \pm 0.12$ |
| SmODE-SAC | $767.13 \pm 14.25$ | $0.18 \pm 0.00$ | $0.71 \pm 0.01$ | $1.09 \pm 0.02$ | $1.22 \pm 0.02$ | $2.39 \pm 0.06$ | $2.72 \pm 0.16$ | $1.84 \pm 0.04$ |
| ActionChunk-SAC | $734.25 \pm 33.78$ | $0.39 \pm 0.01$ | $1.04 \pm 0.04$ | $2.16 \pm 0.05$ | $1.34 \pm 0.09$ | $2.78 \pm 0.17$ | $5.64 \pm 2.11$ | $2.54 \pm 0.11$ |
| DWS-SAC (Ours) | $\mathbf{796.90 \pm 4.11}$ | $\mathbf{0.12 \pm 0.00}$ | $\mathbf{0.42 \pm 0.00}$ | $\mathbf{0.70 \pm 0.01}$ | $\mathbf{0.42 \pm 0.01}$ | $\mathbf{0.83 \pm 0.03}$ | $\mathbf{1.51 \pm 0.10}$ | $\mathbf{0.91 \pm 0.02}$ |

*Table 15.* **Performance comparison on the DMC Walker task.** Mean$\pm$std over 5 seeds. Higher reward is better; all smoothness-related metrics are lower-is-better. Best results within each backbone block (TD3 or SAC) are highlighted in **bold**.

| Algorithm | Reward$\uparrow$ | Smooth.$\downarrow$ | AFR$_{L2}\downarrow$ | AFR$_{L1}\downarrow$ | Jerk$_{RMS}\downarrow$ | Jerk$_{P95}\downarrow$ | Delta$_{max}\downarrow$ | Delta$_{P95}\downarrow$ |
|---|---|---|---|---|---|---|---|---|
| Vanilla-TD3 | $716.75 \pm 257.45$ | $0.49 \pm 0.09$ | $1.78 \pm 0.31$ | $2.97 \pm 0.53$ | $3.19 \pm 0.52$ | $4.96 \pm 0.71$ | $3.96 \pm 0.35$ | $2.96 \pm 0.41$ |
| L2C2-TD3 | $692.54 \pm 329.86$ | $0.27 \pm 0.07$ | $0.81 \pm 0.22$ | $1.34 \pm 0.40$ | $1.55 \pm 0.39$ | $2.72 \pm 0.73$ | $3.79 \pm 0.54$ | $1.84 \pm 0.50$ |
| ActionChunk-TD3 | $21.62 \pm 10.13$ | - | - | - | - | - | - | - |
| LipsNet++-TD3 | $\mathbf{792.48 \pm 315.01}$ | $0.48 \pm 0.10$ | $1.63 \pm 0.33$ | $2.89 \pm 0.61$ | $3.03 \pm 0.50$ | $4.93 \pm 0.63$ | $3.89 \pm 0.30$ | $2.90 \pm 0.35$ |
| SmODE-TD3 | $661.23 \pm 308.31$ | $0.40 \pm 0.10$ | $1.33 \pm 0.34$ | $2.43 \pm 0.61$ | $2.58 \pm 0.54$ | $4.59 \pm 0.82$ | $4.01 \pm 0.30$ | $2.75 \pm 0.48$ |
| DWS-TD3 (Ours) | $786.24 \pm 332.30$ | $\mathbf{0.22 \pm 0.06}$ | $\mathbf{0.78 \pm 0.19}$ | $\mathbf{1.31 \pm 0.37}$ | $\mathbf{1.41 \pm 0.36}$ | $\mathbf{2.63 \pm 0.68}$ | $\mathbf{3.35 \pm 0.43}$ | $\mathbf{1.79 \pm 0.40}$ |
| Vanilla-SAC | $513.98 \pm 214.74$ | $0.41 \pm 0.09$ | $1.32 \pm 0.25$ | $2.47 \pm 0.54$ | $2.46 \pm 0.42$ | $4.14 \pm 0.46$ | $3.44 \pm 0.29$ | $2.44 \pm 0.25$ |
| L2C2-SAC | $622.44 \pm 251.77$ | $0.33 \pm 0.08$ | $1.07 \pm 0.23$ | $1.98 \pm 0.51$ | $1.91 \pm 0.40$ | $3.28 \pm 0.46$ | $3.13 \pm 0.26$ | $2.11 \pm 0.22$ |
| ActionChunk-SAC | $549.18 \pm 232.88$ | $0.35 \pm 0.11$ | $1.18 \pm 0.30$ | $2.11 \pm 0.68$ | $2.19 \pm 0.52$ | $3.80 \pm 0.63$ | $3.41 \pm 0.27$ | $2.36 \pm 0.32$ |
| LipsNet++-SAC | $158.22 \pm 92.82$ | $0.31 \pm 0.06$ | $0.97 \pm 0.19$ | $1.83 \pm 0.37$ | $1.75 \pm 0.33$ | $3.04 \pm 0.50$ | $3.06 \pm 0.41$ | $1.97 \pm 0.35$ |
| SmODE-SAC | $885.91 \pm 201.52$ | $0.47 \pm 0.05$ | $1.51 \pm 0.15$ | $2.83 \pm 0.33$ | $2.82 \pm 0.26$ | $4.72 \pm 0.27$ | $3.71 \pm 0.22$ | $2.76 \pm 0.14$ |
| DWS-SAC (Ours) | $\mathbf{901.51 \pm 188.92}$ | $\mathbf{0.27 \pm 0.05}$ | $\mathbf{0.86 \pm 0.13}$ | $\mathbf{1.64 \pm 0.27}$ | $\mathbf{1.48 \pm 0.24}$ | $\mathbf{2.45 \pm 0.31}$ | $\mathbf{2.75 \pm 0.32}$ | $\mathbf{1.60 \pm 0.16}$ |

### E.1. Simulation Environment and FCEV Model

The LearningEMS environment simulates the longitudinal dynamics and multi-physics energy flow of a fuel cell hybrid electric logistics truck. The powertrain consists of a proton exchange membrane fuel cell stack serving as the primary power source and a lithium-ion battery pack for energy buffering and peak shaving(He et al., 2024; Wu et al., 2023).

**Longitudinal Dynamics.** The power demand is determined by the vehicle's longitudinal kinetics. The total resistance force $F_r$ is calculated as the sum of rolling resistance, aerodynamic drag, and inertial force:

$$F_r = G \cdot f + \frac{1}{2}\rho A_f C_D v^2 + m \cdot acc \tag{18}$$

where $G$ is gravity, $f$ is the rolling resistance coefficient, $\rho$ is air density, $A_f$ is the frontal area, $C_D$ is the aerodynamic drag coefficient, $v$ is velocity, and $m$ is vehicle mass. The power request $P_{req}$ is derived from $P_{req} = F_r \cdot v$.

**Fuel Cell System.** The hydrogen consumption rate $\dot{m}_{H_2}$ is modeled based on efficiency maps derived from experimental data. It is calculated as:

$$\dot{m}_{H_2} = \frac{P_{fcs}}{\eta_{fcs}(P_{fcs}) \cdot LHV_{H_2}} \tag{19}$$

where $P_{fcs}$ is the fuel cell system output power, $\eta_{fcs}$ is the system efficiency, and $LHV_{H_2}$ is the hydrogen low calorific value. The model also accounts for fuel cell degradation, penalizing conditions such as high-power operation and rapid load changes(Wu et al., 2024a; Yang et al., 2023).

**Battery System.** The battery pack is modeled using an equivalent circuit model. The state of charge (SOC) dynamics and terminal voltage $V_t$ are governed by:

$$V_t = V_{oc}(SOC) - I(t)R_0 \tag{20}$$

where $V_{oc}$ is the open-circuit voltage, $I(t)$ is the current, and $R_0$ is the internal resistance.

### E.2. MDP Formulation

We formulate the FCEV energy management problem as a Markov Decision Process (MDP) tuple $(S, A, P, R, \gamma)$. The objective is to minimize the total cost of hydrogen and system degradation while maintaining the battery SOC.

**State Space ($S$).** The state vector $s_t$ captures the vehicle's kinematic status and the internal states of the powertrain:

$$s_{FCEV} = \{v_t, acc_t, SOC_t, P^t_{fcs}\} \tag{21}$$

where $v_t$ is the vehicle speed, $acc_t$ is acceleration, $SOC_t$ is the battery state of charge, and $P^t_{fcs}$ is the current fuel cell power output. Including $P^t_{fcs}$ allows the agent to observe and control power fluctuations.

**Action Space ($A$).** To ensure the durability of the fuel cell stack, the action space is defined as the rate of change in power (power slope) rather than the absolute power level. The action $a_t$ is continuous:

$$a_t = \Delta P_{fcs} \in [-10\text{kW}, 10\text{kW}] \tag{22}$$

The next fuel cell power command is determined by $P^{t+1}_{fcs} = P^t_{fcs} + a_t$.

**Reward Function ($R$).** The reward function is a multi-objective cost function designed to minimize operational costs and prolong component life. The reward $r_{FCEV}$ is defined as:

$$r_{FCEV} = -\left(C_{H_2} + C_{degr} + \alpha\omega(SOC_{ref} - SOC_t)^2\right) \tag{23}$$

Here, $C_{H_2}$ represents the cost of hydrogen consumption, and $C_{degr}$ represents the monetized cost of fuel cell degradation (caused by load changes and high-power operation). The final term enforces a soft constraint to keep the battery SOC near the reference value $SOC_{ref}$, weighted by coefficient $\omega$.

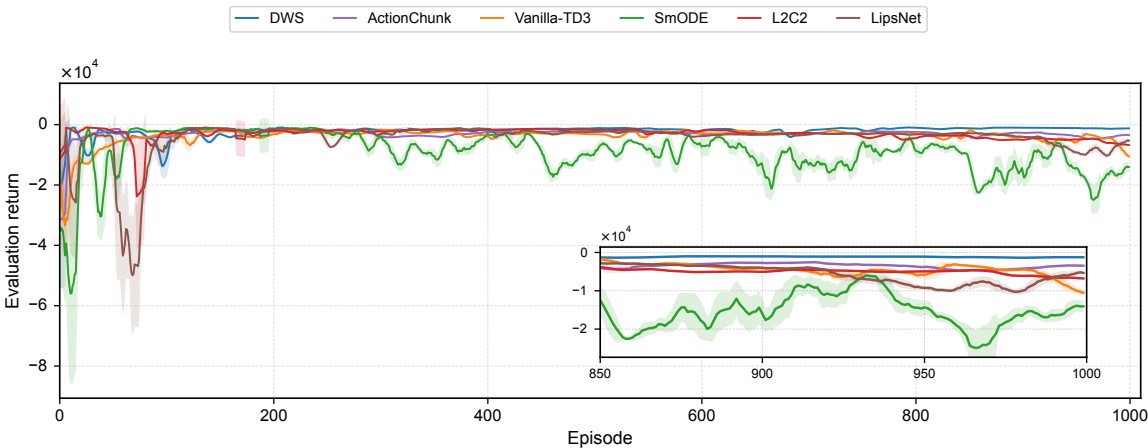

*Figure 14.* Training dynamics on the FCEV energy management task. The curves display the mean evaluation return $\pm$ standard deviation over 1,000 episodes. The zoomed-in inset (bottom right) highlights the asymptotic performance in the final phase, where **DWS** demonstrates superior stability and highest converged return compared to baselines.

## E.3. Training Dynamics and Performance Evaluation

To comprehensively evaluate the learning efficiency and asymptotic performance of our proposed method, we conducted a comparative training experiment spanning 1,000 episodes within the LearningEMS FCEV environment. The primary evaluation metric is the cumulative episodic return, which is defined as the negative sum of hydrogen fuel costs, fuel cell degradation costs, and SOC violation penalties. Consequently, a higher return (closer to zero) indicates a more optimal energy management strategy with lower operational costs and extended component lifespan.

We compare **DWS** against five competitive baselines: ActionChunk, TD3, SmODE, L2C2, and LipsNet. The training curves, illustrated in Figure 14, reveal the following insights:

- **Convergence and Stability:** As shown in the figure, **DWS** (blue line) demonstrates superior sample efficiency, converging rapidly to a high-return policy within the initial 200 episodes. Unlike baselines such as SmODE (green line), which exhibits significant variance and performance collapses—likely due to frequent violations of the power slope constraints and subsequent penalties—DWS maintains a stable trajectory throughout the training process.

- **Asymptotic Performance:** The zoomed-in inset (episodes 850–1000) highlights the asymptotic behavior of the converged policies. DWS consistently achieves the highest evaluation return among all compared methods. This indicates that our method effectively smooths the fuel cell power output, thereby minimizing the degradation cost component ($C_{degr}$) of the reward function while satisfying the load demand.

- **Comparison with Baselines:** While TD3 and ActionChunk show competitive performance, they settle at a slightly lower return compared to DWS, suggesting suboptimal trade-offs between energy conservation and battery SOC maintenance. The significant gap between DWS and the other smoothing-based methods (L2C2, LipsNet) further validates the effectiveness of our approach in the highly non-linear FCEV control landscape.

## F. Additional Experiments on Autonomous Driving Scenarios

We further evaluate **DWS** on safety-critical driving tasks in CARLA. These experiments stress-test temporal smoothness under tight safety constraints, where high-frequency control oscillations can directly trigger lane departures or collisions.

Unless otherwise noted, we report metrics computed on the *executed* actions applied to the environment.

### F.1. Backbone and Compared Methods

**RLHF-style backbone (HG-TD3).** For CARLA, we build upon a TD3-style actor–critic pipeline augmented with (i) **human guidance** (behavioral cloning on intervention-labeled samples) and (ii) **Q-advantage replay** that upweights

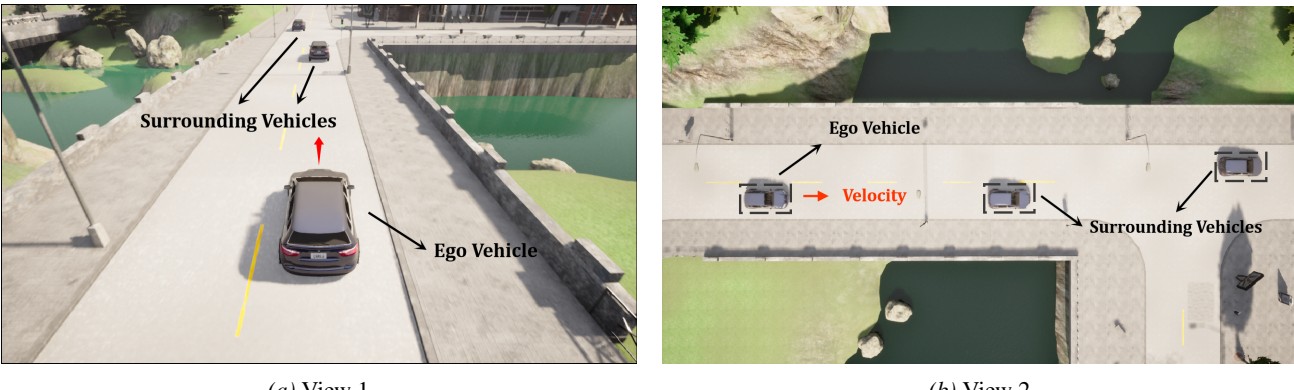

*(a)* View 1                                                    *(b)* View 2

*Figure 15.* **LCO scenario in CARLA.**

informative transitions based on a learned advantage signal. We refer to this common training pipeline as **HG-TD3**. All CARLA methods share the same HG-TD3 backbone and differ *only* by the added action-smoothing component.

**Baselines.**    We compare against representative smoothing approaches integrated into the same backbone: **L2C2**, **ActionChunk**, **LipsNet++**, and **SmODE**, as well as the plain backbone (**HG-TD3**). Our method, **DWS**, couples an actor-side smoothness regularizer with a training-time *value window* (terminal-safe windowed $h$-step auxiliary TD targets constructed from contiguous executed segments), and an optional deployment-time *execution window* that enforces local action coherence under the same horizon $h$.

**Fairness.**    All methods use identical environment interaction budgets, network architectures, and evaluation protocols. We evaluate each trained policy over a fixed number of episodes and report mean±std for continuous metrics. Task-level outcomes (success/collision/lane-departure) are reported as percentages.

### F.2. CARLA: LCO (Lane-change overtaking)

#### F.2.1. SCENARIO AND TERMINATION

The LCO task requires the ego vehicle to overtake two NPC vehicles via lane changes while avoiding collisions and lane departures. Figure 15 illustrates representative camera views.

The ego vehicle follows a predefined road segment with a *narrow drivable lane*, making the task highly sensitive to high-frequency steering jitter. An episode terminates when one of the following conditions is met: (i) **success** (the ego completes the overtaking route and reaches the goal region), (ii) **collision** with any object/vehicle, or (iii) **lane departure** (the ego crosses the lane boundary beyond a tolerance threshold). Because lane departure immediately ends the episode, even small high-frequency steering jitter can cause repeated boundary violations and early termination, yielding low success despite partial progress.

#### F.2.2. STATE AND ACTION SPACES

**State space.**    We formulate the task as an MDP. The state is represented by a semantic segmentation observation from the ego-centric camera. Following common practice, we downsample the segmentation map to reduce computation while preserving driving-relevant structure. Concretely, the state at time $t$ is a single-channel matrix of size $45 \times 80$ with normalized pixel values:

$$\mathbf{s}_t \in [0, 1]^{45 \times 80}. \tag{24}$$

**Action space.**    For LCO, the control objective is primarily lateral, so the action is the **steering command**:

$$a_t \in [-a_{\max}, a_{\max}], \tag{25}$$

where $a_{\max} \in (0, 1]$ scales the maximum steering magnitude. Positive values correspond to right steering and negative values to left steering. Other low-level controls (e.g., throttle/brake) are handled by the scenario controller to focus learning

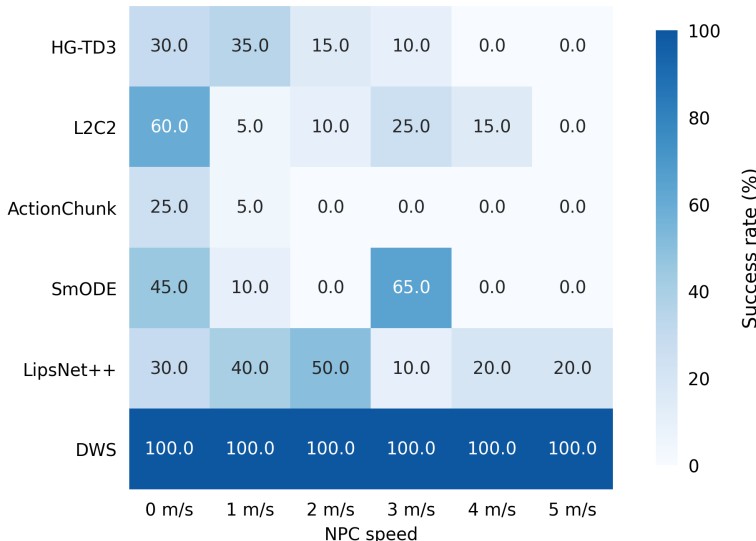

*Figure 16.* **LCO: robustness to NPC speed.** Success rate (%) across NPC speeds $\{0, 1, 2, 3, 4, 5\}\,\mathrm{m/s}$.

*Table 16.* **Dynamic Lane**: SR/CR/BR/PC by NPC speed (V0–V5). SR: success rate, CR: collision rate, BR: beyond-road rate, PC: normalized path completion (max-normalized within each speed). All values are in %.

| | V0 (0 m/s) | | | | V1 (1 m/s) | | | | V2 (2 m/s) | | | | V3 (3 m/s) | | | | V4 (4 m/s) | | | | V5 (5 m/s) | | | |
|---|---|---|---|---|---|---|---|---|---|---|---|---|---|---|---|---|---|---|---|---|---|---|---|---|
| Algorithm | SR↑ | CR↓ | BR↓ | PC↑ | SR↑ | CR↓ | BR↓ | PC↑ | SR↑ | CR↓ | BR↓ | PC↑ | SR↑ | CR↓ | BR↓ | PC↑ | SR↑ | CR↓ | BR↓ | PC↑ | SR↑ | CR↓ | BR↓ | PC↑ |
| HG-TD3 | 30.0 | 35.0 | 35.0 | 83.5 | 35.0 | 45.0 | 20.0 | 88.9 | 15.0 | 30.0 | 55.0 | 61.7 | 10.0 | 35.0 | 55.0 | 60.6 | 0.0 | 100.0 | 0.0 | 62.9 | 0.0 | 75.0 | 25.0 | 68.0 |
| L2C2 | 60.0 | 15.0 | 25.0 | 89.3 | 5.0 | 50.0 | 45.0 | 74.5 | 10.0 | 30.0 | 60.0 | 68.4 | 25.0 | 35.0 | 40.0 | 68.5 | 15.0 | 45.0 | 40.0 | 70.9 | 0.0 | 95.0 | 5.0 | 78.6 |
| ActionChunk | 25.0 | 60.0 | 15.0 | 85.7 | 5.0 | 75.0 | 20.0 | 81.9 | 0.0 | 70.0 | 30.0 | 78.6 | 0.0 | 70.0 | 30.0 | 77.6 | 0.0 | 80.0 | 20.0 | 81.1 | 0.0 | 50.0 | 50.0 | 38.4 |
| SmODE | 45.0 | 55.0 | 0.0 | 87.5 | 10.0 | 40.0 | 50.0 | 87.1 | 0.0 | 80.0 | 20.0 | 77.2 | 65.0 | 35.0 | 0.0 | 94.6 | 0.0 | 85.0 | 15.0 | 84.3 | 0.0 | 100.0 | 0.0 | 87.1 |
| LipsNet++ | 30.0 | 70.0 | 0.0 | 80.4 | 40.0 | 30.0 | 30.0 | 93.7 | 50.0 | 20.0 | 30.0 | 95.9 | 10.0 | 90.0 | 0.0 | 90.9 | 20.0 | 65.0 | 15.0 | 92.5 | 20.0 | 55.0 | 25.0 | 74.3 |
| DWS (Ours) | **100.0** | **0.0** | **0.0** | **100.0** | **100.0** | **0.0** | **0.0** | **100.0** | **100.0** | **0.0** | **0.0** | **100.0** | **100.0** | **0.0** | **0.0** | **100.0** | **100.0** | **0.0** | **0.0** | **100.0** | **100.0** | **0.0** | **0.0** | **100.0** |

on stable lateral control during overtaking.

### F.2.3. NPC SPEED SETTINGS AND ROBUSTNESS SWEEP

Unless otherwise specified, the main setting uses **NPC speed** $5\,\mathrm{m/s}$ (V5), which is the most challenging configuration. To test robustness, we additionally sweep NPC speeds in $\{0, 1, 2, 3, 4, 5\}\,\mathrm{m/s}$ while keeping other environment settings unchanged. Figure 16 reports success rates across speeds for all methods.

**Outcome rates across speeds.** Table 16 reports SR/CR/BR/PC by NPC speed. A key observation is that, under the hardest setting (V5), several baselines exhibit **0% success rate**. This does *not* imply that these methods fail to learn any useful behavior: as shown in Table 16, at easier settings (e.g., V0), baselines can achieve non-trivial success. Moreover, even when SR drops to zero at V5, baselines can still obtain moderate **path completion (PC)**, indicating partial progress before termination. This is consistent with the termination condition: because lane departure ends the episode immediately, a policy that is broadly competent but exhibits persistent high-frequency steering jitter may repeatedly cross the narrow lane boundary during overtaking, causing early termination and thus near-zero SR despite non-trivial PC.

### F.2.4. METRICS

We report both **task outcome** metrics and **comfort/smoothness** metrics. Task performance includes success rate (SR), collision rate (CR), boundary-violation rate (BR), and path completion (PC). Comfort/smoothness metrics quantify temporal oscillations in executed actions and vehicle motion, including action smoothness, yaw smoothness/yaw rate, acceleration statistics (mean and tail percentile), and sideslip frequency. All smoothness metrics are computed from the *executed* controls applied to the environment, ensuring consistent evaluation under windowed execution.

*Table 17.* Comfort metrics on Dynamic Lane (NPC speed V0), reported as mean±std. Lower is better (↓).

| Algorithm | Action smoothness ↓ | Yaw smoothness ↓ | Yaw rate ↓ | AbsAcc mean ↓ | AbsAcc P95 ↓ | Sideslip > 3° ↓ |
|---|---|---|---|---|---|---|
| HG-TD3 | 0.0112±0.0012 | 0.0036±0.0004 | 0.0888±0.0092 | 0.820±0.118 | 5.857±0.539 | 0.647±0.060 |
| L2C2 | 0.0077±0.0028 | 0.0024±0.0010 | 0.0591±0.0240 | 0.643±0.115 | 4.661±0.543 | 0.347±0.162 |
| ActionChunk | 0.0792±0.0150 | 0.0015±0.0003 | 0.0359±0.0071 | 0.563±0.089 | 3.761±0.246 | 0.203±0.043 |
| SmODE | 0.0094±0.0018 | 0.0038±0.0007 | 0.0941±0.0186 | 0.872±0.185 | 5.192±0.709 | 0.630±0.071 |
| LipsNet++ | 0.0108±0.0022 | 0.0034±0.0011 | 0.0843±0.0280 | 1.110±0.134 | 5.296±0.506 | 0.485±0.089 |
| DWS (Ours) | **0.0029±0.0005** | **0.0014±0.0002** | **0.0341±0.0048** | **0.332±0.014** | **2.780±0.247** | **0.115±0.041** |

*Table 18.* Comfort metrics on Dynamic Lane (NPC speed V1), reported as mean±std. Lower is better (↓).

| Algorithm | Action smoothness ↓ | Yaw smoothness ↓ | Yaw rate ↓ | AbsAcc mean ↓ | AbsAcc P95 ↓ | Sideslip > 3° ↓ |
|---|---|---|---|---|---|---|
| HG-TD3 | 0.0077±0.0015 | 0.0024±0.0005 | 0.0609±0.0123 | 0.631±0.084 | 4.330±0.362 | 0.392±0.074 |
| L2C2 | 0.0183±0.0026 | 0.0041±0.0009 | 0.1026±0.0225 | 0.908±0.066 | 6.303±0.375 | 0.672±0.077 |
| ActionChunk | 0.0771±0.0135 | 0.0016±0.0002 | 0.0391±0.0059 | 0.508±0.040 | 3.758±0.266 | 0.187±0.029 |
| SmODE | 0.0041±0.0006 | 0.0021±0.0003 | 0.0528±0.0068 | 0.542±0.084 | 3.923±0.162 | 0.294±0.048 |
| LipsNet++ | 0.0132±0.0015 | 0.0038±0.0006 | 0.0952±0.0141 | 0.905±0.075 | 5.390±0.350 | 0.451±0.070 |
| DWS (Ours) | **0.0032±0.0005** | **0.0015±0.0001** | **0.0378±0.0037** | **0.324±0.014** | **2.547±0.291** | **0.173±0.026** |

**Comfort metrics by speed.** To provide a detailed view beyond the main-table results at V5, Table 17–Table 22 report comfort/smoothness metrics across V0–V5. Overall, DWS consistently reduces action/yaw oscillations and tail-risk motion statistics (e.g., AbsAcc P95 and sideslip frequency) across all traffic speeds.

### F.2.5. TRAINING PROTOCOL AND EVALUATION

Training follows the HG-TD3 pipeline with off-policy replay. All methods are trained for **300 training rounds** under the same interaction budget, network architecture, and optimization hyperparameters (unless otherwise noted). We periodically evaluate the current policy (every fixed number of episodes) and record evaluation return throughout training, as shown in Figure 8. For the final comparison, we evaluate each method over a fixed set of episodes under the main NPC speed (V5, $5\,\mathrm{m/s}$) and report the aggregated statistics in Table 2 in the main paper.

### F.2.6. ADDITIONAL DISCUSSION: WHY SMOOTHNESS MATTERS HERE

LCO places an unusually strong premium on smooth control. The lane is narrow and overtaking requires sustained lateral tracking with minimal margin. If the learned policy exhibits high-frequency steering oscillations, the ego vehicle can "wobble" laterally, repeatedly triggering lane departures and early terminations. This explains why some baselines may show partial progress (e.g., non-trivial PC) yet still achieve near-zero success in the hardest traffic setting: small but persistent action jitter can prevent completing a full overtaking maneuver even when the policy is broadly competent. In contrast, DWS reduces high-frequency oscillations in steering and steering increments ( Figure 17), improving stability and enabling consistent completion without boundary violations.

*Table 19.* Comfort metrics on Dynamic Lane (NPC speed V2), reported as mean±std. Lower is better (↓).

| Algorithm | Action smoothness ↓ | Yaw smoothness ↓ | Yaw rate ↓ | AbsAcc mean ↓ | AbsAcc P95 ↓ | Sideslip $> 3°$ ↓ |
|---|---|---|---|---|---|---|
| HG-TD3 | 0.0048±0.0008 | 0.0017±0.0004 | 0.0432±0.0105 | 0.498±0.042 | 4.019±0.168 | 0.254±0.043 |
| L2C2 | 0.0082±0.0012 | 0.0025±0.0005 | 0.0629±0.0136 | 0.688±0.060 | 4.994±0.368 | 0.438±0.091 |
| ActionChunk | 0.0795±0.0148 | 0.0011±0.0002 | 0.0275±0.0043 | 0.470±0.032 | 3.652±0.245 | 0.173±0.028 |
| SmODE | 0.0093±0.0024 | 0.0028±0.0005 | 0.0693±0.0127 | 0.783±0.096 | 5.172±0.524 | 0.478±0.104 |
| LipsNet++ | 0.0083±0.0008 | 0.0029±0.0003 | 0.0726±0.0068 | 0.774±0.078 | 4.797±0.269 | 0.333±0.034 |
| DWS (Ours) | **0.0010±0.0002** | **0.0006±0.0001** | **0.0145±0.0022** | **0.230±0.004** | **1.035±0.072** | **0.042±0.011** |

*Table 20.* Comfort metrics on Dynamic Lane (NPC speed V3), reported as mean±std. Lower is better (↓).

| Algorithm | Action smoothness ↓ | Yaw smoothness ↓ | Yaw rate ↓ | AbsAcc mean ↓ | AbsAcc P95 ↓ | Sideslip $> 3°$ ↓ |
|---|---|---|---|---|---|---|
| HG-TD3 | 0.0104±0.0020 | 0.0032±0.0006 | 0.0789±0.0141 | 0.774±0.080 | 5.522±0.752 | 0.594±0.111 |
| L2C2 | 0.0087±0.0019 | 0.0023±0.0005 | 0.0564±0.0134 | 0.662±0.089 | 4.818±0.357 | 0.413±0.090 |
| ActionChunk | 0.0833±0.0108 | 0.0012±0.0001 | 0.0292±0.0036 | 0.465±0.025 | 3.625±0.225 | 0.178±0.038 |
| SmODE | 0.0050±0.0017 | 0.0022±0.0004 | 0.0543±0.0091 | 0.534±0.095 | 4.011±0.533 | 0.354±0.072 |
| LipsNet++ | 0.0079±0.0014 | 0.0031±0.0004 | 0.0770±0.0110 | 0.780±0.072 | 4.865±0.481 | 0.395±0.072 |
| DWS (Ours) | **0.0010±0.0001** | **0.0008±0.0001** | **0.0194±0.0023** | **0.208±0.006** | **0.866±0.097** | **0.106±0.058** |

*Table 21.* Comfort metrics on Dynamic Lane (NPC speed V4), reported as mean±std. Lower is better (↓).

| Algorithm | Action smoothness ↓ | Yaw smoothness ↓ | Yaw rate ↓ | AbsAcc mean ↓ | AbsAcc P95 ↓ | Sideslip $> 3°$ ↓ |
|---|---|---|---|---|---|---|
| HG-TD3 | 0.0039±0.0014 | 0.0013±0.0004 | 0.0324±0.0109 | 0.469±0.058 | 3.688±0.313 | 0.215±0.072 |
| L2C2 | 0.0077±0.0007 | 0.0024±0.0003 | 0.0594±0.0079 | 0.607±0.066 | 4.535±0.308 | 0.422±0.088 |
| ActionChunk | 0.0788±0.0096 | 0.0012±0.0002 | 0.0297±0.0051 | 0.435±0.033 | 3.475±0.160 | 0.161±0.042 |
| SmODE | 0.0061±0.0007 | 0.0020±0.0003 | 0.0512±0.0068 | 0.550±0.043 | 3.837±0.256 | 0.306±0.043 |
| LipsNet++ | 0.0106±0.0013 | 0.0037±0.0003 | 0.0915±0.0074 | 0.843±0.054 | 5.246±0.395 | 0.456±0.063 |
| DWS (Ours) | **0.0014±0.0002** | **0.0007±0.0001** | **0.0182±0.0016** | **0.228±0.016** | **1.309±0.215** | **0.081±0.020** |

*Table 22.* Comfort metrics on Dynamic Lane (NPC speed V5), reported as mean±std. Lower is better (↓).

| Algorithm | Action smoothness ↓ | Yaw smoothness ↓ | Yaw rate ↓ | AbsAcc mean ↓ | AbsAcc P95 ↓ | Sideslip $> 3°$ ↓ |
|---|---|---|---|---|---|---|
| HG-TD3 | 0.0068±0.0022 | 0.0023±0.0006 | 0.0580±0.0160 | 0.562±0.095 | 4.084±0.553 | 0.348±0.105 |
| L2C2 | 0.0085±0.0026 | 0.0027±0.0009 | 0.0678±0.0225 | 0.632±0.132 | 4.734±0.887 | 0.428±0.162 |
| ActionChunk | 0.1024±0.0151 | 0.0012±0.0004 | 0.0299±0.0089 | 0.525±0.027 | 4.113±0.171 | 0.315±0.074 |
| SmODE | 0.0072±0.0019 | 0.0027±0.0006 | 0.0667±0.0145 | 0.627±0.097 | 4.504±0.775 | 0.423±0.116 |
| LipsNet++ | 0.0124±0.0019 | 0.0035±0.0008 | 0.0871±0.0196 | 0.833±0.077 | 5.257±0.570 | 0.413±0.087 |
| DWS (Ours) | **0.0014±0.0002** | **0.0007±0.0001** | **0.0187±0.0022** | **0.211±0.015** | **1.195±0.144** | **0.071±0.019** |

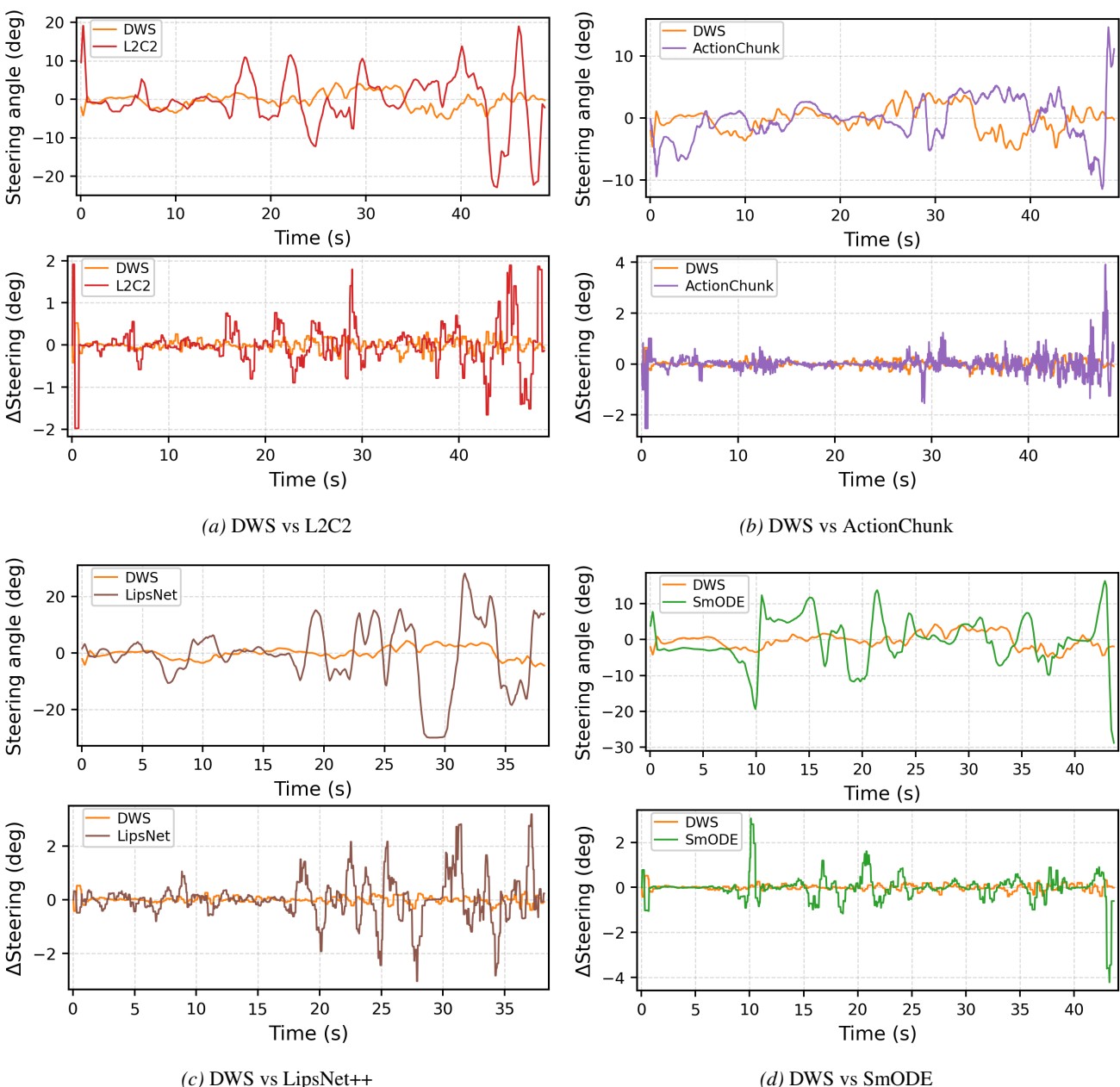

*(a)* DWS vs L2C2

*(b)* DWS vs ActionChunk

*(c)* DWS vs LipsNet++

*(d)* DWS vs SmODE

*Figure 17.* **Pairwise steering trajectories in LCO (appendix).** For each baseline, we show steering angle (top) and steering increment Δsteer (bottom) from representative successful episodes under identical plotting/smoothing settings. Compared to each baseline, DWS consistently suppresses high-frequency oscillations in Δsteer, which reduces lateral "wobbling" and helps avoid boundary-violation terminations in the narrow-lane overtaking route.

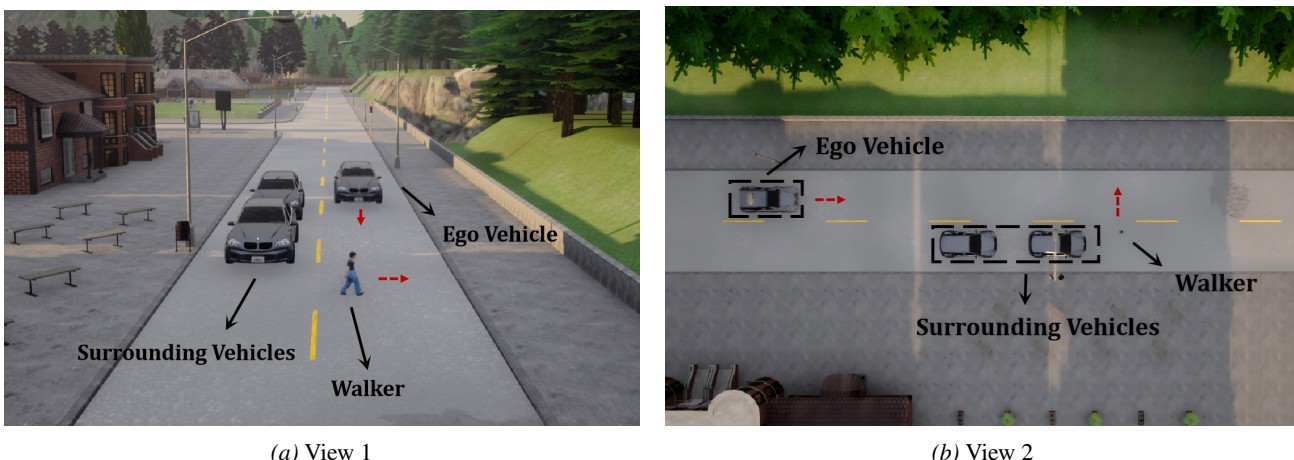

*(a)* View 1        *(b)* View 2

*Figure 18.* **AEB scenario in CARLA.** Representative camera views.

## F.3. CARLA: AEB (Autonomous Emergency Braking)

### F.3.1. SCENARIO AND TERMINATION

We specify the AEB scenario in CARLA Town01 as follows. The ego vehicle starts from a fixed spawn point $(x, y) = (338.5, 190.0)$ and cruises at a target speed of $60\,\mathrm{km/h}$ ($16.7\,\mathrm{m/s}$). A pedestrian is spawned ahead at $(x, y) = (331.0, 260.0)$ and walks laterally with a constant speed of $1.0\,\mathrm{m/s}$ (WalkerControl). Two static obstacle vehicles are also placed ahead at $(x, y) = (335.0, 248.0)$ and $(335.0, 255.0)$.

An episode terminates when any of the following occurs: (i) **collision** (triggered by the collision sensor); (ii) **finish** after the pedestrian has fully passed (defined as pedestrian $x > 341$) and the ego reaches a terminal longitudinal position ($y > 265$); or (iii) reaching the episode horizon. Lane departure is not considered in AEB, as we disable lane-departure termination for this scenario.

### F.3.2. STATE AND ACTION SPACES

**State space.** The observation is a semantic segmentation image from an ego-centric camera. We use the semantic camera output and downsample it from $1280 \times 720$ to $80 \times 45$, then flatten and normalize to $[0, 1]$:

$$\mathbf{s}_t \in [0, 1]^{45 \times 80}. \tag{26}$$

During training, we randomize the camera configuration by sampling the mounting height from $\{1.2, 1.3, 1.4\}$ (scaled by vehicle bounds) and the semantic camera FoV from $\{80°, 90°, 100°\}$ each episode.

**Action space.** The agent outputs a single continuous braking command:

$$a_t \in [0, 1], \tag{27}$$

where larger values correspond to stronger braking intensity. Throttle is controlled by a simple proportional cruise controller to track the target speed when braking is inactive; steering is fixed to $0$ (straight driving).

### F.3.3. METRICS AND ACTIVE BRAKING WINDOW

We report both task outcomes and safety/comfort metrics. Task outcomes include success rate (SR) and collision rate (CR). Safety margin is measured by time-to-collision (TTC) statistics (mean/min), and comfort is measured by $\mathrm{Acc}_{\mathrm{RMS}}$ and $\mathrm{Jerk}_{\mathrm{RMS}}$, all computed from the executed trajectory. $\mathrm{Acc}_{\mathrm{RMS}}$ and $\mathrm{Jerk}_{\mathrm{RMS}}$ are reported in two variants: (i) full-episode statistics, and (ii) statistics restricted to an *active braking window* that focuses on the safety-critical interaction phase.

The active window is defined in the evaluation script as:

$$\text{active} \Leftrightarrow (\neg\texttt{ped\_passed}) \wedge (d_y^{\text{signed}} > 0.5) \wedge (v > 0.5),$$

where $d_y^{\text{signed}} = y_{\text{ped}} - y_{\text{ego}}$ denotes the signed longitudinal distance (positive when the pedestrian is ahead of the ego vehicle), and $v$ is the ego speed in $\text{m/s}$. We estimate acceleration and jerk from finite differences of the logged ego speed using timestep $\Delta t = 1/f$ with $f = 25\,\text{Hz}$: $a_t = (v_t - v_{t-1})/\Delta t$ and $j_t = (a_t - a_{t-1})/\Delta t$. TTC statistics (mean/min) are computed on the same active window.

### F.3.4. TRAINING AND EVALUATION PROTOCOL

All CARLA methods share the same RLHF-style backbone (HG-TD3) with human guidance: when manual intervention is detected (keyboard/steering wheel input), the environment returns a human action and the transition is stored with an intervention flag; otherwise the agent action is stored normally. We evaluate every 5 episodes and summarize training-time curves such as evaluation return and cumulative collision rate. Unless otherwise specified, training runs for **151 episodes** (set by maximum_episode) with identical environment settings and evaluation protocol across methods.

### F.3.5. LEARNING CURVES FOR COMFORT AND SAFETY MARGIN

**Additional quantitative results.** Table 23 reports outcome, safety margin, and comfort metrics for AEB (mean±std). All methods achieve identical task performance (100% success and 0% collision), so the comparison focuses on *how* braking is executed. DWS achieves the **highest** $\text{TTC}_{\text{mean}}$ on the active braking window (1.791 vs. 1.327–1.708), indicating a larger safety margin during the critical interaction phase. Meanwhile, DWS substantially improves braking comfort by reducing both $\text{Acc}_{\text{RMS}}$ and $\text{Jerk}_{\text{RMS}}$ on the active window (1.386 and 47.900) compared to the best baseline (2.633 and 90.284). These gains also persist on the full-episode metrics, where DWS yields the lowest $\text{Acc}_{\text{RMS}}$ and $\text{Jerk}_{\text{RMS}}$ overall (0.836 and 28.921), suggesting consistently smoother longitudinal control beyond the active braking interval.

We further visualize training/evaluation dynamics in Figure 20.

Following our protocol, we run a noise-free evaluation every 5 episodes and log (i) comfort via Jerk RMS ( Figure 20a) and (ii) safety margin via the mean TTC on the active braking window ( Figure 20b). We also plot the training-phase smoothness loss (Smoothness MSE) in Figure 20c, which is most informative during optimization as it quickly saturates near zero in evaluation.

*Table 23.* **AEB evaluation summary (expanded).** Mean±std over test episodes. All metrics are computed on executed actions. Metrics with "active" are computed on the active braking window.

| Algorithm | SR (%) | CR (%) | $\text{TTC}_{\text{mean}}$ active↑ | $\text{Acc}_{\text{RMS}}$ active↓ | $\text{Acc}_{\text{RMS}}$↓ | $\text{Jerk}_{\text{RMS}}$ active↓ | $\text{Jerk}_{\text{RMS}}$↓ |
|---|---|---|---|---|---|---|---|
| HG-TD3 | 100.0±0.0 | 0.0±0.0 | 1.708±0.108 | 2.770±0.072 | 1.487±0.026 | 97.527±2.567 | 52.338±0.947 |
| L2C2 | 100.0±0.0 | 0.0±0.0 | 1.677±0.084 | 2.819±0.131 | 1.502±0.036 | 99.013±4.531 | 52.783±1.271 |
| ActionChunk | 100.0±0.0 | 0.0±0.0 | 1.667±0.069 | 1.431±0.083 | 1.516±0.028 | 94.273±2.945 | 50.438±1.018 |
| LipsNet++ | 100.0±0.0 | 0.0±0.0 | 1.682±0.052 | 2.633±0.129 | 1.434±0.036 | 90.284±4.692 | 49.268±1.450 |
| SmODE | 100.0±0.0 | 0.0±0.0 | 1.327±0.131 | 2.669±0.114 | 1.547±0.035 | 93.958±3.976 | 54.467±1.227 |
| DWS(Ours) | 100.0±0.0 | 0.0±0.0 | **1.791±0.075** | **1.386±0.128** | **0.836±0.037** | **47.900±4.656** | **28.921±1.391** |

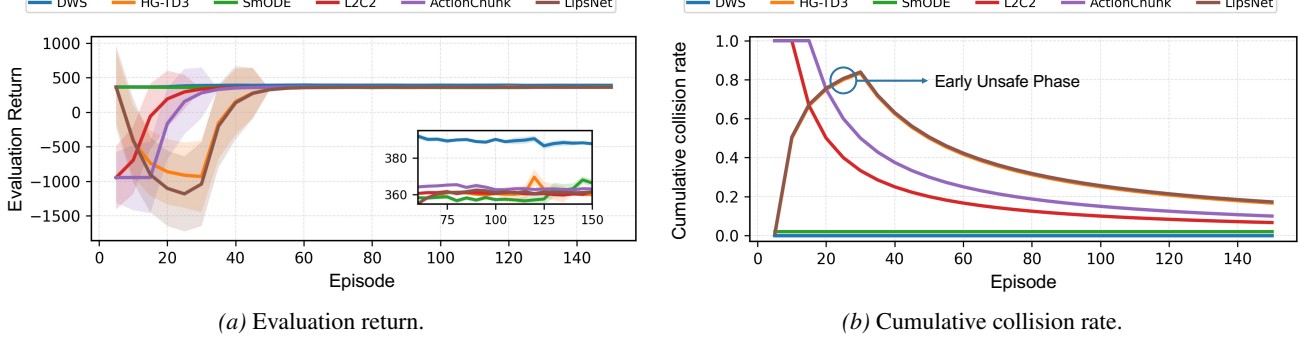

*(a)* Evaluation return.          *(b)* Cumulative collision rate.

*Figure 19.* **AEB training-time evaluation curves (every 5 episodes).**

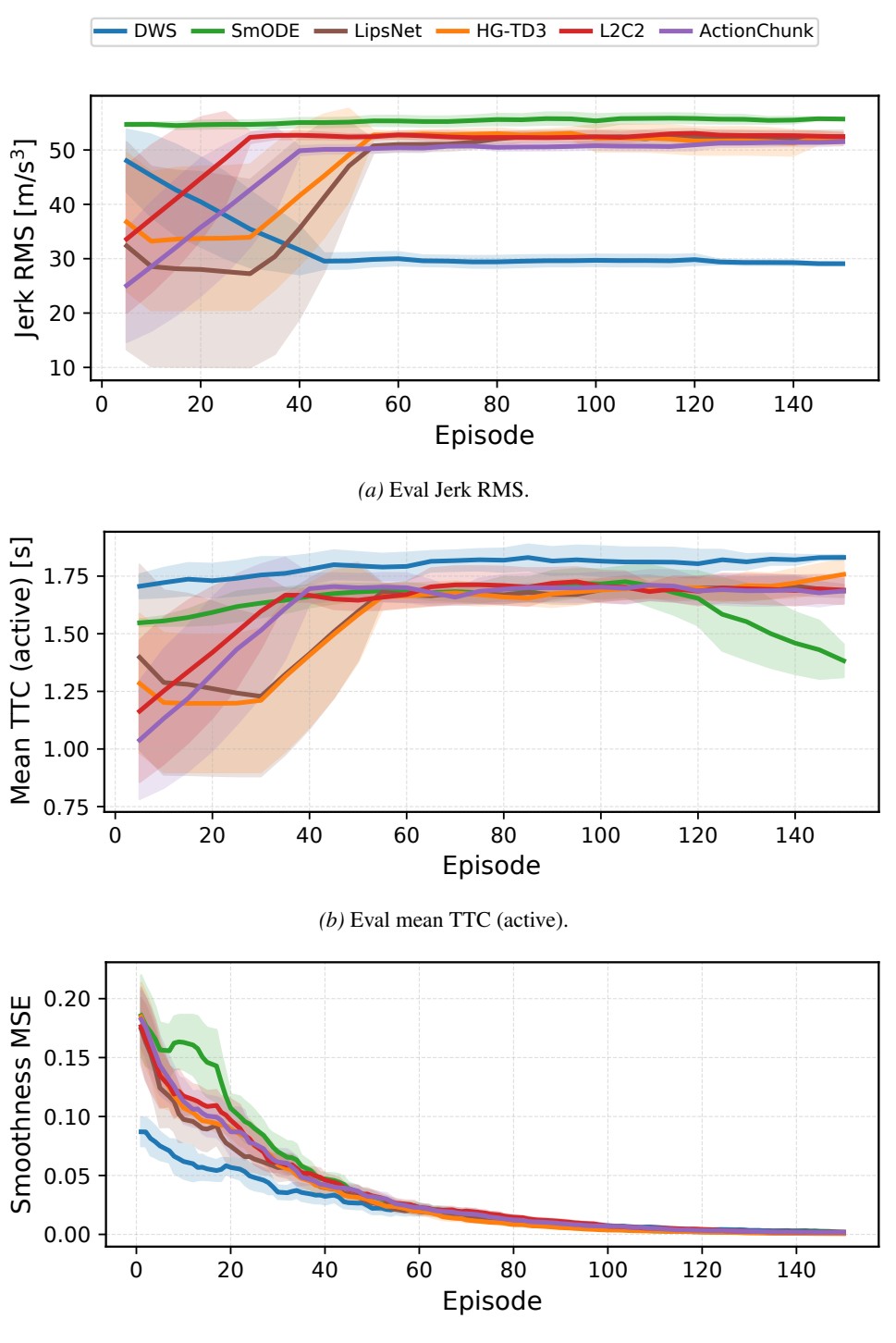

*(a)* Eval Jerk RMS.

*(b)* Eval mean TTC (active).

*(c)* Train Smoothness MSE.

*Figure 20.* **AEB learning curves.** Noise-free evaluation every 5 episodes; shaded regions show variability.

