# OpenReview forum: "Implicit Action Chunking for Smooth Continuous Control"
_ICML.cc/2026/Conference — ICML 2026 regular_

### Official Review · Reviewer_iQNt · 2026-03-05

**Soundness:** 2
**Presentation:** 3
**Significance:** 2
**Originality:** 3
**Overall Recommendation:** 4
**Confidence:** 4

**Summary:**

This paper introduces a new action chunking paradigm that seeks to overcome previous challenges that are fundamental to explicit action chunking such as optimization difficulties due to expanded horizon length and incompatibility with prior one-step RL algorithms. The method introduced to combat these issues is called Dual-Window Smoothing (DWS), which sidesteps issues of explicit methods, i.e. methods that expand the action selection horizon for the policy from one-step to multi-step to enforce temporal coherence, by instead using an implicit scheme. This implicit scheme avoids expanding the action selection horizon by instead using a standard one-step policy alongside a deterministic operator which maps that one-step action to a multi-step action trajectory and then uses a pair of windows to more seamlessly integrate this design with standard RL frameworks. DWS then is evaluated across three regimes of continuous control and shows promise across all three.

**Compliance With Llm Reviewing Policy:**

Affirmed.

**Final Justification:**

The authors have addressed my concern regarding the evaluation suite by expanding their DMC benchmarking to match LipsNet++. While I maintain that these tasks are relatively 'easy' in terms of sample efficiency, the authors have shown that their evaluation is now in line with recent work in this area. Given the alignment with prior work further validating their method I will raise my score to a weak accept.

**Key Questions For Authors:**

- Shouldn’t F_h be conditioned on not only the action of the policy, but also the state, s_t? The action alone implies that action sequences are state independent which seems quite wrong given that an action chunked policy was conditioned on state-action pairs as is the reactive policy for traditional RL. Originally I thought that you calling the a_t the latent action somehow implied that it was encoding state-action and then being transformed into the physical action u_t, but given that w_k transforming a_t to u_t is simply a scalar/linear transformation, my original question about conditioning remains.
- A follow on to the above question: what makes a_t a latent action if it is simply a scaled down version of the physical action via w_t? Furthermore, throughout this paper you call a_t a number of different names including an atomic action and a latent action. The multitude of labels add to confusion, not help solve it.
- Why were the first two benchmarks selected? Carla makes sense to me, but to my points in the weaknesses section, I am not seeing a particularly good motivating reason for selection of "easy" DMC tasks or the FCEV task. The lack of substantial assessment of this method more broadly across "harder" continuous control tasks is a major factor in my evaluation of this paper being low.

**Limitations:**

I don't see any reference to limitations in their work. The authors simply talk about their own work. Placing some sort of limitations that lead into future directions in the conclusion would have been a good place for this.

**Strengths And Weaknesses:**

**Strengths:**
- The concept of the idea is timely and useful as action chunking has become more commonplace
- Of particular note, having the temporal coherence without requiring as much engineering to plug in the method to standard interfaces is an important consideration that has been in part ignored by prior work
- The change to a value window to address the myopic concern when making action commitments over a chunk is a nice, albeit complicated way to solve this problem
- Great results in the Carla section. On almost all metrics and across both the AEB and LCO tasks the DWS method performs best, which certainly indicates it works well in this regime.

**Weaknesses:**
- In point 3 of the problems you introduce in the introduction you discuss the idea of conventional safety shielding and fallback mechanisms without citation. Further lead in needs to be done if you are going to talk about action chunking breaking these methods since there is no context given for them.
- You talk about how prior action chunking methods add complication, but the description you give of DWS sounds much more complicated than the storage and indexing required to integrate with actor-critic backbones and replay buffers. There are certainly other issues, which you cover later in the paper and in part when introducing prior work in the intro, but the introduction of those problems need to be more explicitly mentioned and connected when you talk about "complicating integration with established actor-critic backbones."
- I think in (3) of your contributions in the intro it would be good to compare the 100% success rate to what prior methods could do. If most prior methods were achieving 80% success that says something very different than if they were achieving 99% success previously. As presented now it is hard to discriminate the significance of your work.
- In the pseudocode appendix highlighting unchanged portions of the algorithm would be helpful to show how easy it is to add DWS and further clarify what is novel in a visual way.
- While the usage of Carla with raw RGB mitigates some of my concerns about the utility of this for high dimensional continuous control, the rationale around picking the DMC control environments as provided in appendix D isn’t convincing. Drawing connections between these simple environments and autonomous driving subtasks seems spurious at best given that 1. based on the framing and title this is a paper about continuous control more broadly, not the task of autonomous driving and 2. the driving subtasks need to be accomplished in parallel not in isolation so proving your method works on very simple surrogates does not contribute to the soundness of your approach.
- The DMC environments chosen by the authors are by far the easiest environments to solve in that benchmark and the space used in this paper would be better spent on harder DMC environments if the goal is to prove this method is useful broadly for continuous control. Furthermore, were the DMC environment observations proprioceptive or image based? It was not clear in the paper or appendix and should be specified when the environments are introduced.
- Likewise to the choice of DMC environments, the choice of FCEV as a baseline is strange to me for a paper with framing around continuous control more broadly. From what I have seen (and based on the citation counts) this is not a well-studied benchmark in the RL literature and it is quite low dimensional from what I inferred from the appendix. As a result it seems like another “toy” example that is useful to see in a short blurb or in the appendix, but not one that helps set this method apart as a winner for smoothing continuous control more broadly since it is relatively niche and simple.
- Overall this paper seems to have an issue of being framed as a general RL method for smoothing RL for general continuous control, but that frame is broken by poor choice of benchmarks and motivation for their selection. In both the final sections of the paper and the appendix the work seems continually reframed to reference autonomous vehicles, which detracts from the messaging of the algorithm. If this method is truly meant to be a replacement for action chunking across general continuous control then it needs to be tested across a more general and well known set of benchmarks.

---

> ### Author Rebuttal · Authors · 2026-03-30
>
> We thank the reviewer for the careful reading and detailed feedback. To address concerns regarding benchmark coverage and task complexity, we supplement our original evaluation **with three additional challenging tasks: DMC Cheetah, CityLearn [1], and CARLA Town02.** We provide detailed responses to each concern through the five sections below.
>
> **1. Introduction Clarity. (W1, W2, W3)**
> We thank the reviewer for this comment. We will add citations for safety shielding and fallback mechanisms, clarifying that explicit chunking’s open-loop execution prevents point-wise expert override critical for safety-critical RL.
>
> The introduction did not clearly explain the specific integration challenges of prior action chunking methods, such as storing and indexing multi-step actions in actor-critic backbones and replay buffers. While discussed later in the paper, we will revise the introduction to explicitly highlight these difficulties and show that DWS achieves action smoothing without extra buffer or indexing overhead, simplifying integration with standard actor-critic architectures.
>
> We will also revise the introduction to present this result in comparative context. In the hardest LCO setting, DWS achieves 100% success versus 20% for the strongest baseline and 0% for others, demonstrating substantial robustness and reliability.
>
> **2. Pseudocode and Visual Clarity. (W4)**
> We will revise the pseudocode in Appendix A.2 to visually highlight unchanged standard actor-critic components versus the minimal DWS additions. This will demonstrate the plug-and-play nature of the integration and visually clarify the architectural novelty.
>
> **3. State Conditioning and Terminology. (Q1, Q2)**
> We clarify that state dependence is introduced through the policy as $a_t = \pi(s_t)$. The function  $F_h$  then deterministically maps this state conditioned reference action into an executed action profile. Within each window, the executed actions are generated from the reference action computed at the boundary state, rather than being state independent open loop outputs. Therefore, the executed sequence is not state independent. Instead, the augmented state formulation in Appendix A.2 preserves the Markov property by including the within window phase and the cached reference action. We will also unify the terminology, consistently refer to $a_t$ as the reference action and removing inconsistent labels such as latent or atomic.
>
> **4. Benchmark Selection and Additional Challenging Tasks. (Q3, W5, W6, W7, W8)**
> We thank the reviewer for raising this important concern. Our benchmark selection covers **standard RL validation, real-world industrial control, and high-dimensional perception tasks.**
>
> DMC is a standard benchmark widely used in continuous control and is also adopted by LipsNet++ (ICML 2025 Spotlight), ensuring fair comparison with prior action-smoothing methods. SmODE (ICLR 2025 Spotlight) is evaluated on MuJoCo, however, since DMC is itself built on MuJoCo, our benchmark choice remains closely aligned with prior work.
>
> FCEV represents a real industrial control problem with tightly coupled subsystems (motor, battery, fuel cell, and vehicle dynamics). Though state/action dimensions are moderate, the system exhibits high complexity and stringent physical constraints. This exemplifies industrial control tasks that are low-dimensional yet deployment-critical, contrasting with high-dimensional simulation benchmarks. With tens of millions of hybrid electric vehicles deployed globally, successful RL deployment in EMS offers substantial economic benefits and emission reduction. Our goal is to demonstrate that RL can extend beyond simulation to deliver practical value in real-world applications.
>
> **To further validate our method under more challenging conditions, we evaluate on three additional tasks:**   (1) DMC Cheetah, a more demanding continuous-control task requiring dynamic balance and locomotion. (2) CityLearn, a recent benchmark for energy-flexible, resilient, occupant-centric, and carbon-aware management of grid-interactive communities. (3) CARLA Town02, a high-dimensional autonomous driving scenario in which the method is extensively tested across ten routes with joint throttle and steering control using raw RGB inputs. These additions validate that DWS generalizes from mechanism-level validation to complex, high-dimensional, and safety-critical applications, addressing concerns about task simplicity. **Full tables: https://anonymous.4open.science/r/63e7/README.md.**
>
> **5. Limitations**
> We will add a dedicated Limitations section in the conclusion. Full details are provided in our response to Reviewer eELD.
>
> Finally, we sincerely thank the reviewer for the careful reading and constructive feedback.
>
> > Reference:
> [1] Nweye et al. "CityLearn v2: Energy-flexible, resilient, occupant-centric, and carbon-aware management of grid-interactive communities," Journal of Building Performance Simulation, 2025.

---

> > ### Author Rebuttal · Reviewer_iQNt · 2026-04-03
> >
> > Thank you to the authors for clarifying weaknesses 1-3, offering to revise the pseudocode in weakness 4, and for your explanations of my key questions.
> >
> > I agree with the authors that DMC is a standard benchmark widely used in continuous control contexts. I did not claim to the contrary. My problem was not with the benchmark, but with the chosen tasks, the minimal explanation for why they were selected, and the actual difficulty (as measured by samples required to solve) for these environments. I think the reasoning of ensuring fair comparison by comparing to prior work in the same benchmarks makes sense, but then why not compare only to Lipsnet in cart pole, reacher, cheetah, and walker instead of the overly simple tasks you chose. That would match what they did in evaluation in DMC. Furthermore, just because prior work had insufficiently difficult benchmark choices does not sway my opinion. Likewise, SmODE deployed in walker and humanoid tasks, which again are much harder (at least humanoid is) than those selected for evaluation in this paper.
> >
> > Under the author's response to benchmark selection it is noted that you chose DMC to ensure fair comparison to prior methods since it is widely used. Your admission of this fact makes the selection of FCEV even more confusing to me. If anything I think renaming the section titles for the DMC and Carla results to remove "case study", moving the FCEV section to be 4.2, and expanding the DMC evaluations to include DMC walker run at the absolute least (to match LipsNet++'s evaluation tasks fully) makes the most sense and better ensures that not only "easy" tasks are used for testing. My preference would be to deploy it in humanoid and quadruped tasks in DMC too. For the FCEV results I believe the case study title still makes sense because it is a seemingly arbitrary choice that is not common in the literature when evaluating action chunking methods. While I appreciate the author's efforts to demonstrate real-world utility, these additions don't address my core concern about the method's performance on high dimensional standard benchmarks that the RL community measures progress with. Given my issues with FCEV I have the same issues with adding CityLearn. This is not a widely used benchmark in the literature and its inclusion after rebuttal does not make sense from the perspective of choosing benchmarks that allow fair comparison to prior work.
> >
> > With all that being said I appreciate the additional experiment of adding DMC's half cheetah to the benchmarks you test your algorithm on and the addition of a further CARLA task.
> >
> > I will raise my score in appreciation of the authors improvements to the paper, but I still have reservations about this work centering around the difficulty and generality of evaluation.

---

> > > ### Author Response · Authors · 2026-04-07
> > >
> > > **1. Addition of Harder DMC Benchmarks**
> > >
> > > We sincerely thank the reviewer for the constructive and insightful feedback.  We agree this is a key issue to address. We have added results on **DMC Walker-run, together with the previously included DMC Cheetah task**. These are more challenging locomotion benchmarks. DWS remains strong under both backbones when compared with standard baselines. For clarity, we summarize the SAC results below.
> > >
> > > DMC Cheetah (SAC backbone)
> > >
> > > | Method             | Reward ↑           | Smooth ↓        |
> > > |--------------------|------------------:|----------------:|
> > > | Vanilla-SAC        | 777.35 ± 24.84    | 0.41 ± 0.01     |
> > > | L2C2-SAC           | 432.90 ± 183.74   | 0.13 ± 0.04     |
> > > | LipsNet++-SAC      | 347.98 ± 15.83    | 0.40 ± 0.00     |
> > > | SmODE-SAC          | 767.13 ± 14.25    | 0.18 ± 0.00     |
> > > | ActionChunk-SAC    | 734.25 ± 33.78    | 0.39 ± 0.01     |
> > > | **DWS-SAC (Ours)**     | **796.90 ± 4.11**     | **0.12 ± 0.00**     |
> > >
> > > DMC Walker-run (SAC backbone)
> > >
> > > | Method             | Reward ↑           | Smooth ↓        |
> > > |--------------------|------------------:|----------------:|
> > > | Vanilla-SAC        | 513.98 ± 214.74   | 0.41 ± 0.09     |
> > > | L2C2-SAC           | 622.44 ± 251.77   | 0.33 ± 0.08     |
> > > | LipsNet++-SAC      | 158.22 ± 92.82    | 0.31 ± 0.06     |
> > > | SmODE-SAC          | 885.91 ± 201.52   | 0.47 ± 0.05     |
> > > | ActionChunk-SAC    | 549.18 ± 232.88   | 0.35 ± 0.11     |
> > > | **DWS-SAC (Ours)**     | **901.51 ± 188.92**   | **0.27 ± 0.05**     |
> > >
> > > DWS achieves the best return and the best smoothness across all methods on both tasks. Together with the Cheetah results, this strengthens the evaluation beyond easier settings and provides a more convincing comparison on harder locomotion benchmarks. **Full tables including all DMC tasks are available at
> > > https://anonymous.4open.science/r/fff7/README.md**
> > >
> > >
> > > **2. Benchmark Selection and Generality**
> > >
> > > We appreciate your thoughtful guidance regarding benchmark selection. This is indeed a very interesting point. Our work is originally motivated by real world control problems. Action smoothness is critical for safety and system longevity in those settings. This is why we initially focused on autonomous driving and industrial control.
> > >
> > > We agree that evaluating only a limited subset of DMC tasks does not fully demonstrate generality. Following the reviewer’s suggestion, we expand our evaluation to **seven DMC tasks**, including Cartpole, Reacher, Cheetah, and Walker.  With this addition, our evaluation now **fully matches the DMC task suite used in LipsNet++**, ensuring a fair and complete comparison with prior action smoothing methods. Across all tasks, DWS consistently achieves strong performance in both return and smoothness, indicating good generalization across different control regimes and difficulty levels. Humanoid and quadruped tasks are indeed valuable for further validation. They exhibit locomotion characteristics similar to Walker and Cheetah, such as balance control and coordinated motion over long horizons. In the revision, we will further clarify the evaluation structure to make this alignment more explicit.
> > >
> > > In summary, the updated evaluation now includes harder locomotion benchmarks, aligns with prior work in task coverage, and demonstrates consistent improvements in both performance and smoothness across all considered tasks.
> > >
> > > We thank the reviewer again for helping us significantly strengthen the completeness of our empirical evaluation.

---

### Official Review · Reviewer_eELD · 2026-03-12

**Soundness:** 3
**Presentation:** 3
**Significance:** 3
**Originality:** 3
**Overall Recommendation:** 5
**Confidence:** 4

**Summary:**

This paper proposes an implicit action chunking method that prescribes action evolution within a specified window. To achieve this, it introduces a dual-window smoothing mechanism that performs deterministic modulation while aligning temporal-difference targets. The approach is empirically evaluated across diverse and challenging domains, including the DeepMind Control Suite, an industrial energy management task, and vision-based autonomous driving.

**Compliance With Llm Reviewing Policy:**

Affirmed.

**Final Justification:**

The paper is overall technically solid and the rebuttal addressed my concerns. Therefore, I maintain my assessment.

**Key Questions For Authors:**

- Given that the map is fixed, to what extent does this sacrifice the expressiveness and adaptability of the learned policy?

**Limitations:**

No.

Suggestions:
- Discuss the potential restriction on policy expressiveness and adaptability

**Strengths And Weaknesses:**

Strengths:

- The proposed solution is conceptually simple yet empirically powerful, avoiding unnecessary algorithmic complexity.
- The method is supported by strong experimental results demonstrating its effectiveness across a wide variety of simulated and real-world benchmarks.
- The paper features clear, high-quality illustrations that effectively communicate the core mechanism to the reader.

Weaknesses:

- Relying on a fixed map schedule may restrict the policy's ability to adapt to highly dynamic or unpredictable states, potentially capping the model's overall capacity.

---

> ### Author Rebuttal · Authors · 2026-03-30
>
> We sincerely thank the reviewer for the positive evaluation and for raising this important and insightful question regarding policy expressiveness under a fixed execution schedule.
>
> **1. Fixed Schedule and Expressiveness. (W1, Q1)**
> We thank the reviewer for the comment. We agree that the fixed execution schedule introduces a structural constraint that theoretically limits expressiveness compared to fully unconstrained step-wise control or learned modulation. However, this is an intentional inductive bias. It encourages smooth and physically feasible control trajectories that respect real-world actuator constraints.
>
> Expressiveness is preserved at the decision level. The policy remains fully state-conditional and replans at every window boundary with high frequency, using the default window length h = 3. This allows the agent to adapt rapidly to state changes while the fixed intra-window modulation ensures smooth interpolation and avoids high-frequency jitter.
>
> Our sensitivity analysis shows that performance is robust across a range of window lengths, indicating that the fixed schedule does not severely impair adaptability in the evaluated regimes. We acknowledge that this may be a limitation in highly dynamic scenarios that require intra-window adaptation. Future work will explore state-dependent or adaptive execution profiles. These could include learnable window lengths or modulation schedules that adjust dynamically based on state uncertainty or task dynamics, while retaining the stability benefits of structured execution. **Full tables are provided at: https://anonymous.4open.science/r/5fda/README.md**
>
> **2. Limitation and future direction. (Suggestion)**
> We will add a dedicated discussion noting that fixed execution profiles may limit adaptability in highly dynamic environments. Future directions include state-dependent execution profiles and adaptive window lengths to balance reactivity and temporal coherence.
>
> **Limitations and Future Work:**
> While DWS demonstrates strong empirical performance across diverse domains, it currently relies on a fixed execution horizon in the execution window. This introduces a structural inductive bias that may limit policy expressiveness and adaptability in highly dynamic or unpredictable environments, particularly under abrupt changes within a window. A key direction for future work is to relax this fixed-schedule assumption through state-dependent or learnable execution profiles, adaptive window lengths, or hybrid schemes that balance step-wise reactivity with temporally structured execution.  Additionally, we will extend the implicit action chunking framework to offline RL architectures, leveraging its compatibility with standard replay buffers for large-scale pre-training. Finally, validation on complex robotic manipulation tasks with high-dimensional actions and contact-rich dynamics will further demonstrate the generality and practical deployment potential of this approach.
>
> Finally,  we thank the reviewer for these insightful questions regarding policy expressiveness.

---

> > ### Author Rebuttal · Reviewer_eELD · 2026-04-02
> >
> > Thanks for the rebuttal which has fully resolved my concerns. I maintain my positive assessment.

---

### Official Review · Reviewer_5C65 · 2026-03-13

**Soundness:** 3
**Presentation:** 3
**Significance:** 3
**Originality:** 3
**Overall Recommendation:** 5
**Confidence:** 4

**Summary:**

This paper studies smooth continuous control in actor-critic RL and proposes Dual-Window Smoothing (DWS), which keeps the policy output dimension unchanged but executes each policy output through a short deterministic window, trains the critic with a matched horizon-aligned value target, and adds a first-order action-difference regularizer. Experiments cover five DeepMind Control Suite tasks, an EV energy-management benchmark, and two CARLA driving scenarios built on a human-guided TD3 pipeline.

**Compliance With Llm Reviewing Policy:**

Affirmed.

**Final Justification:**

Thank you to the authors for the detailed rebuttal and clarifications. The rebuttal addressed my main concerns, particularly around experimental details, sensitivity analysis, uncertainty reporting, and clarification of the CARLA setup and baseline comparisons. These responses improved my confidence in the paper’s soundness and clarity.

Overall, I find the paper technically solid, clearly presented, and relevant to an important problem in smooth continuous control. While I still view the originality as primarily a strong integration of existing ideas rather than a fundamentally new paradigm, I believe the contribution is meaningful and likely to be useful to others. On balance, the rebuttal strengthened my assessment, and I have updated my recommendation accordingly.

**Key Questions For Authors:**

1. Why are Q-Chunking / Decoupled Q-Chunking not included as baselines, given that the paper cites them as close prior work and DQC in particular is directly motivated by the open-loop/reactivity problem? A convincing comparison or careful explanation would materially affect my assessment of originality and soundness.

2. Please report the missing uncertainty details for EMS and CARLA: how many independent training seeds were run, how many evaluation episodes underlie Table 2 / Table 11, and whether the headline CARLA results are reproducible across seeds. Strong multi-seed evidence would improve my soundness assessment.

3. Please reconcile the mismatch between the main-text CARLA description and the appendix: are these experiments raw-RGB end-to-end driving with steering and acceleration, or semantic-segmentation inputs with steering-only / braking-only control? This matters because it changes how broadly I interpret the driving claims.

4. How sensitive is DWS to the choice of window length `h`, the execution profile (ZOH vs. decay), and the smoothness weight `lambda_S`? If the method only works in a narrow regime, that would materially weaken the practical contribution.

5. Please provide the missing architecture details for the actor, critic, and vision encoder used in the CARLA experiments, as well as parameter counts and compute budget. This would improve reproducibility and confidence in the reported results.

**Limitations:**

No. The paper includes an impact statement, but it does not adequately discuss technical limitations such as the narrowed driving setup, the lack of seed-based uncertainty for EMS/CARLA, the dependence on semantic segmentation and human guidance in CARLA, and the sensitivity that may arise from the fixed horizon / execution-profile choices.

**Strengths And Weaknesses:**

### Strengths

- The paper addresses a real problem. In physical control settings, action jitter is not a cosmetic issue; it can directly break feasibility and safety.
- The method is conceptually coherent. The execution window, value-window target, and boundary regularizer fit together as a plausible way to smooth execution while keeping a standard actor-critic interface.
- The presentation is generally clear. I found the paper easy to follow at a high level, and the appendix does include pseudocode, the main hyperparameters, and a component ablation.
- The empirical scope is broader than a single-domain demo. The paper does test DWS on standard control benchmarks, an industrial-style EMS task, and two CARLA scenarios.
- The ablation in Table 4 is directionally useful: it suggests the full method is stronger than using only the execution window or only the value-window / regularization pieces.

### Weaknesses

- The autonomous-driving framing is materially overstated. In the main text, the CARLA setup is described as mapping "raw RGB camera inputs" to "continuous steering and acceleration commands." In the appendix, the actual setup is much narrower: both tasks use 45x80 semantic-segmentation inputs, LCO controls steering only while throttle/brake are handled by a scenario controller, and AEB outputs braking only with steering fixed to zero. That is not the same as end-to-end raw-vision driving, so the driving claims need to be weakened substantially.
- Several headline claims are stronger than the evidence. The paper repeatedly claims to outperform SOTA baselines and "consistently" outperform baselines, but the DMC tables do not support that literally. For example, on Reacher-Easy with SAC, SmODE-SAC reports higher reward than DWS-SAC (984.65 vs. 979.75), and on Cartpole-Swingup with SAC, L2C2-SAC reports higher reward than DWS-SAC (863.94 vs. 857.73). On other tasks, the reward differences are very small relative to the reported standard deviations. The experiments support that DWS is competitive and often smoother, not that it uniformly dominates.
- The uncertainty reporting is too weak for the strength of the empirical claims. DMC uses 5 seeds, which is good, but EMS Table 1 reports no variance at all, CARLA Table 2 / Table 11 report task-level percentages without uncertainty, and the paper does not clearly report multi-seed training results for EMS or CARLA. A fixed-episode success rate of 100% in one setup is not enough to justify broad wording like "significant improvement," "SOTA," or "perfect success" without stronger uncertainty estimates.
- Baseline coverage is not strong enough relative to the novelty framing. The closest explicit chunking papers are Q-Chunking and especially Decoupled Q-Chunking, which the paper itself cites as a close advancement addressing open-loop/reactivity issues. However, the experiments only compare against an in-house "ActionChunk" instantiation rather than these stronger prior formulations. That makes the novelty positioning and the "better than explicit chunking" claim less convincing than the paper suggests.
- The CARLA comparison is not fully clean for ActionChunk. The paper states that all baselines except ActionChunk use the same human-guided mechanism. Since the driving results are one of the main pillars of the paper, the ActionChunk comparison partly conflates chunking quality with compatibility with the specific human-guided training pipeline.
- Ablation coverage is incomplete. Table 4 is only on the easier LCO setting at NPC speed 0 m/s, not the main 5 m/s setting, and I did not find sensitivity analysis for the window length `h`, the execution profile choice (ZOH vs. decay), or the smoothness weight `lambda_S`. Since these are central design choices, I would expect at least one such analysis.
- Reproducibility is still below the bar I would want, especially for the vision experiments. The paper does not clearly specify the actor/critic network architectures, the image encoder details, parameter counts, or compute budget. For CARLA and EMS, I also could not find clear seed counts for training runs. Given how central the driving results are, these omissions matter.
- The originality claim should be narrower. The most defensible novelty here is the specific combination of deterministic execution smoothing, a horizon-aligned critic target, and a boundary regularizer inside a standard actor-critic loop. That is a useful combination, but it is not obviously a fundamentally new action-chunking paradigm relative to recent chunked-critic / chunked-action RL papers.

---

> ### Author Rebuttal · Authors · 2026-03-30
>
> We thank the reviewer for the detailed assessment.
>
> **1. CARLA setup. (W1, Q3)**
> We apologize for the confusion. We clarify that the RL policy operates on semantic representations rather than raw RGB inputs. The raw camera observations are first processed by a segmentation module and converted into compact state representations of size 45 by 80, which are then used as inputs to the policy network. This design reduces perceptual complexity and is commonly adopted in vision-based autonomous driving pipelines. We will revise the manuscript to clarify this point.
>
> We further clarify the control design rationale: LCO employs steering only lateral control, while AEB uses braking only longitudinal control. These configurations are intentionally selected to isolate and validate smoothing effects on specific control dimensions, specifically steering for overtaking maneuvers and braking for emergency stops.
>
> To demonstrate full end to end capabilities with joint control, we add experiments on Town02 over 10 routes with simultaneous throttle and steering control, where DWS-SAC achieves superior driving performance and smoother control. **Full results: https://anonymous.4open.science/r/4ff0.**
>
> **2. Strength of empirical claims. (W2)**
> We acknowledge that claims of consistently outperforming SOTA are strong. Our method targets action smoothing in high-dimensional control tasks. While SmODE and LipsNet++ perform strongly on low-dimensional inputs, DWS remains competitive on DMC, with comparable returns and improved smoothness. Minor reward differences are small relative to variance and accompanied by better smoothness metrics. In contrast, DWS significantly outperforms baselines on EMS and CARLA. We will revise the manuscript accordingly.
>
> **3. Uncertainty reporting for EMS/CARLA. (W3, Q2)**
> We add multi seed results for EMS and CARLA. EMS is reported over 5 seeds with 20 evaluation episodes per seed. CARLA LCO  is reported over 5 seeds with 20 episodes per seed. The original CARLA tables use 300 evaluation episodes per condition. The multi seed results preserve the same qualitative ranking and support the stability of DWS. **Full results: https://anonymous.4open.science/r/8cc8/README.md**
>
> **4. Baseline coverage: Q-Chunking. (W4, Q1)**
> As clarified in Appendix B, our ActionChunk baseline is a controlled instantiation of the Q-Chunking (QC) paradigm adapted for our online setting. Original QC targets offline-to-online RL, we modified it to operate in a purely online actor-critic framework and named it ActionChunk. DQC improves upon QC by decoupling the chunk size of the policy from that of the critic. We will clarify this scope in the revision.
>
> **5. ActionChunk fairness under human guidance. (W5)**
> We clarify that ActionChunk uses the same HG-style replay intervention mechanism as other methods. The distinction lies in compatibility. Explicit chunking outputs fixed-length action sequences, so point-wise human overrides disrupt chunk consistency and weaken training efficiency. In contrast, DWS preserves step-wise interaction and supports instantaneous intervention without breaking the execution structure. We will revise the misleading wording.
>
> **6. Ablation, sensitivity, and execution profile. (W6, Q4)**
> We thank the reviewer for this suggestion. We extend the component ablation from the easier LCO setting (NPC speed 0 m/s) to the main challenging setting (5 m/s). The same conclusion holds: each component contributes, while the full DWS achieves the best performance.
>
> We add sensitivity analyses for window length (h) and smoothness weight ($\lambda_S$) on DMC Reacher-Easy and LearningEMS. Results show DWS performs robustly across a wide range and does not rely on narrow tuning.
>
> We compare execution profiles on AEB. Both ZOH and dissipative decay achieve competitive results, indicating DWS is not tied to a specific schedule. ZOH shows slightly better performance.
>
> **Full results:  https://anonymous.4open.science/r/57ae**
>
> **7. Reproducibility details. (W7, Q5)**
> We acknowledge these omissions and will provide complete implementation details in the revision. **Architecture and training details link: https://anonymous.4open.science/r/fd37.** To ensure full reproducibility, we will release all code, data, and detailed hyperparameter configurations upon acceptance.
>
> **8. Originality and limitations. (W8)**
> We respectfully clarify that DWS introduces Implicit Action Chunking, distinct from prior explicit methods. DWS introduces Implicit Action Chunking by unifying deterministic execution smoothing, horizon-aligned value learning, and smoothness regularization within a standard actor-critic loop. This improves temporal coherence without increasing action dimensionality and preserves point-wise intervention capability. Details are provided in our response to Reviewer RieN.
>
> For limitations, full details are provided in our response to Reviewer eELD.
>
> Finally, we thank the reviewer for these insightful.

---

> > ### Author Rebuttal · Reviewer_5C65 · 2026-04-04
> >
> > Thank you to the authors for the detailed rebuttal and additional clarifications. My concerns have been fully addressed, and I have updated my recommendation accordingly.

---

### Official Review · Reviewer_RieN · 2026-03-13

**Soundness:** 3
**Presentation:** 4
**Significance:** 3
**Originality:** 2
**Overall Recommendation:** 4
**Confidence:** 3

**Summary:**

The paper proposes Dual-Window Smoothing (DWS), a method to improve smooth control in reinforcement learning for continuous actions. Standard RL policies make decisions step-by-step, which often leads to high-frequency oscillations and unstable control signals, making them difficult to use in real physical systems. To address this, the authors introduce implicit action chunking, where the policy still outputs a normal action at each step, but the executed actions are smoothed over a short time window.
The method uses two components: an Execution Window, which generates smooth action sequences from a single policy output, and a Value Window, which aligns critic training with the multi-step execution horizon to avoid learning bias. Additionally, a temporal regularization term encourages continuity between consecutive actions. Experiments on several benchmarks (DeepMind Control Suite, energy management, and autonomous driving tasks) show that DWS produces smoother control signals and better performance compared to existing reinforcement learning methods.

**Compliance With Llm Reviewing Policy:**

Affirmed.

**Final Justification:**

I thank the authors for their detailed rebuttal and the additional experiments provided during the discussion period.
The rebuttal addressed my specific technical questions well. The added sensitivity analysis for the window size $h$ and the smoothness regularization weight $\lambda_S​$ on DMC Reacher-Easy and LearningEMS demonstrates that DWS operates robustly within a reasonable parameter range and does not rely on brittle hand-tuning, which was my main concern regarding technical validation.

The clarification of the CARLA reward functions, including the distinction between LCO (with smoothness-aware terms) and AEB (TTC/safety-driven without jerk penalties), satisfactorily answers my question about whether a jerk penalty alone could explain the observed smoothness.
The authors also expanded the LCO evaluation to a broader set of scenarios on Town02 with joint throttle and steering control, which strengthens the generalization claims beyond the originally reported setting.

Weighing the strengths and weaknesses: the paper is well presented, technically sound, and tackles a practical and important problem for deploying RL in physical systems. The dual-window design is a coherent and simple modification that integrates cleanly with standard actor-critic algorithms, and the empirical evaluation spans a meaningful range of domains. On the other hand, the originality remains, in my view, primarily a creative integration of existing ideas (action smoothing, multi-step value targets, temporal regularization) rather than a fundamentally new paradigm. The authors' clarification regarding implicit action chunking as a distinct mechanism from explicit chunking and policy-level smoothing approaches is reasonable, but in my assessment this does not elevate the contribution beyond a strong and useful combination of known concepts.
Overall, the rebuttal reinforced rather than changed my prior assessment. While the rebuttal clears up the technical concerns, the contribution still looks more like a solid integration of existing ideas than a genuinely new direction. I therefore maintain my recommendation: the paper is technically solid and useful to the community working on smooth continuous control, but the contribution, while well-executed, is incremental in nature.

**Key Questions For Authors:**

It would be helpful to know whether different values for the window size h and the smoothness regularization weight $\lambda_S$ were tested. How sensitive is the performance of DWS to these hyperparameters?

Does the LCO experiment consider only a single scenario? If this is the case, it would be valuable to evaluate the method on a broader range of lane-changing scenarios to assess robustness and generalization. Furthermore, could the authors clarify the reward function used in CARLA? In particular, would a reward term penalizing jerk already encourage sufficiently smooth behavior, potentially reducing the need for the proposed smoothing mechanism?

**Limitations:**

Yes

**Strengths And Weaknesses:**

The paper addresses the problem of unstable and oscillatory control signals in reinforcement learning, which often arise because standard RL algorithms make decisions in a step-wise manner while real-world systems require temporally smooth control. The proposed method, Dual-Window Smoothing (DWS), introduces an implicit action chunking mechanism that combines an execution window for smoothing actions during execution and a value window that aligns critic learning with the effective execution horizon.

Overall, the approach is technically sound: the method is clearly defined, the theoretical arguments about intra-window smoothness are reasonable, and the empirical evaluation covers several domains, including control benchmarks, energy management, and autonomous driving. However, the technical validation could be strengthened by a more thorough sensitivity analysis of key hyperparameters such as the window size and smoothness weight.

The paper is generally well structured and clearly motivates the mismatch between step-wise RL decisions and the need for smooth actuation in physical systems. The figures and method description help clarify the dual-window design and its integration with standard actor critic algorithms. However, some experimental details, such as reward functions and justification for certain parameter choices could be described more explicitly.

Regarding significance, the paper tackles an important and practical problem in reinforcement learning for control tasks. Achieving smooth and stable control signals is crucial for applications such as robotics, energy systems, and autonomous driving, and the proposed approach offers a relatively simple modification that can be integrated into existing RL frameworks.

Finally, the originality of the work lies mainly in the combination of existing ideas rather than a completely new paradigm. The concept of smoothing actions and using multi-step value targets is not entirely new, but the proposed implicit action chunking formulation and the coupling of execution smoothing with horizon-aligned value learning represent a creative and coherent integration of these ideas.

---

> ### Author Rebuttal · Authors · 2026-03-30
>
> We sincerely thank the reviewer for the positive assessment and constructive suggestions. We address your concerns regarding sensitivity analysis, scenario diversity, reward design, and originality below.
>
> **1. Sensitivity to key hyperparameters. (W1, Q1)**
> We agree that sensitivity analysis is important. We add new experiments on DMC Reacher-Easy and LearningEMS for both the window size $h$ and the smoothness weight $\lambda_S$. These results show that DWS operates robustly within a reasonable parameter range rather than relying on brittle hand-tuning, validating our default choices.
>
> **Sensitivity to window size $h$:**
> | $h$  | DMC Reward | DMC Smooth |  EMS Reward  |    Cost    |
> | :--: | :--------: | :---------: | :----------: | :--------: |
> |  1   |   783.56   |    0.29     |   -2179.64   |   166.99   |
> |  2   |   893.08   |    0.17     |   -2088.91   |   159.16   |
> |  3   | **922.65** |    0.12     | **-1393.88** | **157.47** |
> |  4   |   873.76   |    0.12     |   -1636.07   |   161.29   |
> |  5   |   845.00   |  **0.11**   |   -2387.92   |   177.40   |
>
> Moderate values ($h=3$, $\lambda_S=0.10$) achieve the optimal reward-smoothness trade-off. Very small $h$ under-utilizes temporal smoothing, while overly large $h$ reduces reactivity. Similarly, too small $\lambda_S$ provides limited regularization benefit, whereas too large $\lambda_S$ over-smooths the policy and suppresses necessary control changes. These results confirm that our default hyperparameters are robust rather than brittle. We will incorporate these sensitivity results into the manuscript revision. **Full sensitivity tables (anonymous): https://anonymous.4open.science/r/0ef5/README.md.**
>
> **2. CARLA reward clarification and experimental details. (W2, Q3)**
> We appreciate this important question. For fairness, all methods use the same task reward. In LCO, the reward already includes smoothness-aware terms together with lane/safety/terminal rewards. In AEB, however, the final reward is mainly TTC/safety-driven and excludes jerk penalties, because adding them made braking overly conservative and reduced stopping performance.
>
> Our main observation is that DWS maintains strong task performance and smoothness under both reward settings, while baseline methods are much more sensitive to the presence or modification of smoothness terms. This suggests that DWS is effective not simply because of reward shaping, but because it addresses the mismatch between step-wise RL decisions and temporally committed actuation through execution smoothing and horizon-aligned value learning. We will clarify the reward definitions in the revision and **provide the full reward formulations in the appendix and anonymous link :** **https://anonymous.4open.science/r/d5fe.**  We will also open-source the code upon acceptance.
>
> **3. LCO scenario diversity. (Q2)**
> Our LCO evaluation is **not** limited to a single scenario. **As reported in Appendix Table 11 and Figure 15** of original submission, we already evaluate LCO across NPC speeds $\{0,1,2,3,4,5\}$ m/s (V0–V5), while the main text uses V5 (hardest) as the representative setting. DWS maintains a **100% success rate** across all tested NPC-speed settings, whereas baseline performance deteriorates substantially as the scenario becomes harder (e.g., HG-TD3 drops from 30% SR at V0 to 0% at V5; LipsNet++ drops from 50% at V2 to 20% at V5).
>
> To further strengthen generalization claims, we add a more complete CARLA evaluation on **Town02 over 10 diverse routes** with joint throttle and steering control, where DWS-SAC shows stronger overall driving performance and smoother control than SAC. **Full Town02 tables are provided in an anonymous table link: https://anonymous.4open.science/r/4ff0/README.md.**
>
> **4. Originality**
> We respectfully clarify the novelty of our proposed DWS, which centers on Implicit Action Chunking (IAC). Unlike existing explicit chunking methods that expand the action space to high-dimensional trajectories, IAC induces temporal coherence without increasing action dimensionality. This is achieved through a dual-window design that combines deterministic execution smoothing with horizon-aligned value learning. Such a design directly addresses the mismatch between step-wise RL optimization and temporally constrained actuation, while preserving standard interfaces and supporting expert intervention.
>
> In contrast to existing action-smoothing methods (e.g., Lipschitz-constrained policies and ODE-based approaches), which mainly operate at the policy or dynamics level and are typically validated on low-dimensional inputs, DWS introduces a structural temporal abstraction mechanism that naturally extends to high-dimensional, perception-driven tasks. To our knowledge, this implicit temporal abstraction, realized without explicit trajectory prediction, represents a distinct paradigm from prior work.
>
> Finally, we sincerely thank you for taking the time out of your busy schedule to review our paper.

---

> > ### Author Rebuttal · Reviewer_RieN · 2026-04-04
> >
> > I thank the authors for the rebuttal which addressed the questions that I had. However, due to the limited novelty I will maintain my score as is.

---

> > > ### Author Response · Authors · 2026-04-07
> > >
> > > We sincerely thank you for confirming that your technical concerns are fully resolved. We deeply appreciate your time and your constructive feedback.
> > >
> > > We respectfully ask you to view our novelty through the unified formulation of our method. Our contribution is not merely smoothing actions or using multiple step targets in isolation. It is the introduction of implicit action chunking. Prior single step methods like LipsNet++ and SmODE primarily improve smoothness through policy network design. Explicit action chunking methods expand the action space to predict open loop trajectories. DWS is distinctly different. We couple a deterministic execution window with a horizon aligned value window. This creates a new temporal abstraction mechanism.
> > >
> > > We believe DWS represents a distinct and practically useful paradigm for the reinforcement learning community. It improves both smoothness and performance without heavy engineering. We hope this highlights our strong practical contribution. We would be very grateful if you might reconsider your score, given that the technical concerns have been fully addressed.

---

### Decision · Program_Chairs · 2026-04-30

**Decision:**

Accept (regular)

**Comment:**

This paper proposes Dual-Window Smoothing, an implicit action chunking framework that improves smooth continuous control by coupling a deterministic execution window with a horizon-aligned value window for temporal-difference targets and an actor-side temporal smoothness regularizer based on first-order action differences.   A feature of this approach is that it does not expand the policy output dimension.

Reviewers agreed that the method is technically sound, clearly presented, and addresses a practical and important problem for deploying reinforcement learning-based policies on physical systems.  Evaluations, especially post rebuttal, span a wide range of domains including autonomous driving tasks, and convincingly demonstrate the benefits of the approach. T

The main point of contention was whether the contribution constitutes a genuinely new paradigm or a coherent integration of existing ideas.  While reviewers didn’t fully agree on this question, all ultimately recognized the contributions as being sufficiently novel to support acceptance. The rebuttal substantially addressed additional concerns about benchmark difficulty, experimental framing, uncertainty reporting, and sensitivity analysis, including expanded evaluation on harder DMC locomotion tasks and multi-seed results for EMS and CARLA.

In summary, the paper presents results which are likely to be useful to the continuous reinforcement learning and control community. We therefore recommend acceptance.